# Cytoplasmic NAD/H synthesis via NRK1 regulates inflammatory capacity and promotes survival of CD4+ T cells

Victoria Stavrou [1,2], Myah Ali[1,2], Nancy Gudgeon [1,2,3], Emma L. Bishop[1,2], Taylor Fulton-Ward[1,2], Bethany Turley[1,2], Silke Heising [2], Sally H. Mohamed[1,4], Sofia Hain [1,4], Lorna George[1,4], Minghao Deng[5], Jack McCowan [6,7], Lozan Sheriff [1,4], Scott P. Davies [1], Bryan Marzullo [2], Daniel A. Tennant[2], Craig L. Doig [5], David A. Bending [1], Ed W. Roberts [6,7], Gareth G. Lavery[5], Rebecca A. Drummond [1,4] & Sarah Dimeloe [1,2] ✉

T cell metabolism increases upon activation, underpinning immune effector functions. Nicotinamide adenine dinucleotide (NAD/H) is an essential redox cofactor for glycolysis and mitochondrial substrate oxidation. It's phosphorylation to NADP/H regulates reactive oxygen species (ROS) abundance. NAD/H levels increase upon T cell activation, but synthesis pathways and implications are not fully characterised. Here, we interrogate the role of the NAD/H-synthesis enzyme nicotinamide riboside kinase 1 (NRK1), the expression of which increases upon stimulation of both human and murine CD4+ T cells. Functionally, NRK1 activity restrains activation and cytokine production of CD4+ T cells while promoting survival. These activities are linked to increased NRK1 expression in the cytoplasm, where it locally raises NAD/H levels. This supports glycolysis, but more profoundly impacts cytoplasmic NADP/H generation, thereby controlling ROS abundance and nuclear NFAT translocation. During fungal and viral infection, T-cell-intrinsic NRK1 maintains effector CD4+ T cell abundance within affected tissues and draining lymph nodes, supporting infection control. Taken together, these data confirm that subcellular regulation of immune cell metabolism determines immune responses at the level of whole organism.

T cell function is underpinned by dynamic changes in metabolism upon antigen encounter. Substantial increases in glycolysis provide biosynthetic precursors for clonal expansion and promote cytokine expression through post-transcriptional regulation. In parallel, elevated glucose and glutamine oxidation in the mitochondrial tricarboxylic acid (TCA) cycle drives electron transport chain (ETC) activity, generating heightened ATP and reactive oxygen species (ROS). ROS are disseminated and signal, promoting T cell activity and differentiation, for example by stabilising nuclear factor of activated T cells (NFAT), which promotes its nuclear translocation[1], but must also be mitigated to prevent oxidative damage.

[1]Department of Immunology and Immunotherapy, College of Medicine and Health, University of Birmingham, Birmingham, UK. [2]Department of Metabolism and Systems Science, College of Medicine and Health, University of Birmingham, Birmingham, UK. [3]Institute of Cancer and Genomic Sciences, College of Medical and Dental Sciences, University of Birmingham, Birmingham, UK. [4]Institute of Microbiology and Infection, College of Medical and Dental Sciences, University of Birmingham, Birmingham, UK. [5]Department of Biosciences, School of Science and Technology, Nottingham Trent University, Nottingham, UK. [6]Cancer Research UK Scotland Institute, Glasgow, UK. [7]School of Cancer Sciences, University of Glasgow, Glasgow, Scotland, UK. ✉e-mail: s.k.dimeloe@bham.ac.uk

Nicotinamide adenine dinucleotide (NAD/H, referring to the total pool) is an essential metabolic redox cofactor, with NAD+ accepting electrons during glycolysis and TCA activity, and conversely, NADH donating electrons to drive ETC activity and lactate production from pyruvate. All cells require continuous NAD/H synthesis, since in addition to its role as a redox cofactor, NAD+ is consumed as a co-substrate for sirtuin (SIRT) and poly-ADP-ribose polymerase (PARP) enzymes, which mediate deacylase and poly (ADP-ribose) polymerase activity, respectively. NAD synthesis pathways include de novo production from tryptophan (kynurenine pathway) or nicotinic acid (Preiss–Handler pathway), or alternatively, salvage of NAD breakdown products. For example, nicotinamide adenine mononucleotide (NAM), generated from NAD by SIRT and PARP activity, is converted to nicotinamide mononucleotide (NMN) and then NAD via the activity of NAM phosphoribosyl-transferase (NAMPT) and NMN adenylate transferases (NMNAT1-3), respectively.

Nicotinamide riboside kinases 1 and 2 (NRK1/2), which were more recently characterised, phosphorylate nicotinamide riboside (NR) into NMN, thereby directing it into the salvage pathway[2–4]. NR is considered a dietary NAD precursor, since it is detected within milk, and oral provision of NR effectively elevates circulating, tissue and cellular NAD/H levels[2,5,6]. However, NMN can also dephosphorylate to generate NR, meaning NRK activity may augment cellular NAD/H independently of NR provision[3]. NRK1 and 2 demonstrate differential expression patterns, with NRK1 being ubiquitously expressed across multiple tissues, and NRK2 largely restricted to skeletal muscle[7]. NRK1 activity appears important within the liver, since NRK1 deficiency in mouse models leads to reduced hepatocyte PARP activity and increased DNA damage, associated with hepatic steatosis and metabolic-associated liver disease[8]. NR supplementation is also reported to promote haematopoiesis in mice, linked to expansion of haematopoietic progenitors, associated with mitophagy and asymmetric mitochondrial distribution upon division[9].

Consistent with heightened metabolic activity, T cells increase intracellular NAD/H abundance upon activation[10,11]. Increased NAD salvage via NAMPT importantly contributes, since NAMPT inhibition decreases cellular NAD/H levels, proliferation and cytokine expression, associated with mitochondrial depolarisation and ATP depletion[10–13]. Additionally, NAMPT inhibitor administration ameliorates the progression of graft-versus-host-disease[12] and experimental autoimmune encephalitis (EAE), a T cell-driven model of multiple sclerosis[10]. Beyond NAMPT, T cell deletion of the NADase CD38 was also shown to increase NAD/H levels, mitochondrial OXPHOS capacity, cytokine expression and anti-tumour function, indicating CD38 also regulates T cell NAD/H status[14]. Finally, it was recently reported that de novo synthesis from tryptophan contributes to NAD/H levels in CD8+ T cells. Specifically, pharmacological or genetic inhibition of this pathway decreased CD8+ T cell NAD/H levels, cytokine expression and anti-tumour function, accompanied by impaired mitochondrial activity and glycolysis[15].

As yet, a role for NRK1 within T cells, or the immune system more generally, is not well defined, however, several studies indicate immunoregulatory potential. For example, NR supplementation in healthy volunteers led to decreases in circulating levels of cytokines, including interleukin (IL)−2, IL-5, IL-6 and tumour necrosis factor-alpha (TNF-α)[5], and similar findings were reported in a cohort of patients with Parkinson's Disease[16]. At the cellular level, analysis of peripheral blood monocytes after NR supplementation in healthy volunteers and individuals with systemic lupus erythematosus confirmed elevated intracellular NAD abundance, associated with downregulation of pro-inflammatory, type I interferon (IFN)-related genes, linked to suppressed type I IFN signalling[6]. Two additional studies have probed the effects of NR supplementation on T cell biology. Firstly, in a murine sepsis model, NR administration was shown to increase T cell NAD/H levels and promote their proliferation and survival, leading to increased frequencies of T helper (Th)1 and Th2 CD4+ T cells. Consistent with this, bacterial load was decreased, associated with decreased organ damage and improved survival[17]. In the second study, in a healthy human cohort, prior NR supplementation was conversely shown to reduce CD4+ T cell differentiation into pro-inflammatory Th1 and Th17 cells upon subsequent stimulation, highlighting the capacity to both promote and suppress T cell activity in different contexts. In the latter study, effects were also observed upon exposure of CD4+ T cells from healthy donors or patients with psoriasis to NR in vitro, and were associated with decreased ROS abundance, linked to induction of anti-oxidant genes and arginosuccinate lyase actvity[18]. Taken together, these studies indicate significant potential for NR to influence T cell biology.

In the current study, the hypothesis that NR and NRK1 activity regulate T cell function was interrogated through comprehensive immune and metabolic analyses of human primary CD4+ T cells and NRK1-deficient murine CD4+ T cells in vitro and in vivo, permitting confirmation of the precise role of NRK1 in these cells and its relevance during infection. The results indicate that generation of NAD/H via NRK1 activity, particularly in the cytoplasm of activated T cells, controls ROS abundance and signalling to regulate CD4+ T cell inflammatory differentiation and function, and promote longevity of these cells during immune responses.

## Results

### NRK1 is upregulated upon human CD4+ T cell activation via CD3 and CD28 signalling; NRK2 is not expressed

T cells substantially increase their metabolic activity upon antigenic stimulation. To begin to probe whether NRK enzyme isoforms may support this, we first assessed their expression in human CD4+ T cells at rest and upon stimulation via CD3 and CD28. Initial quantitative PCR (qPCR) analysis identified that NRK1 transcript abundance was significantly increased in cells stimulated for 48 h compared to unstimulated cells (Supplementary Fig. 1A). NRK2 transcripts were not detected, consistent with established greater tissue restriction of this isoform[7] and in agreement with public datasets reporting no expression in immune cells[19]. We therefore focused our analysis on NRK1 and next assessed protein abundance. Western blot and flow cytometry analysis confirmed that this increased across a time course of CD4+ T cell activation (Fig. 1A, B, gating strategy Supplementary Fig. 1B, C). NRK1 protein was slightly increased in cells activated for 24 h compared to quiescent controls, and reached a peak at 48 h, which was maintained at 72 h. Consistent with these kinetics, analysis in combination with the T cell activation markers, CD25 and CD69 indicated that NRK1 expression was elevated in cells expressing CD25, either alone or together with CD69, but not CD69 in isolation, which identifies early activated cells (Supplementary Fig. 1D). To interrogate signalling pathways driving NRK1 upregulation, contribution of CD3 and CD28 was first assessed by stimulating cells with agonistic antibodies separately or in combination. This identified that CD3 stimulation alone increased NRK1 abundance compared to unstimulated cells, but that this increased further with combined CD3/CD28 stimulation. CD28 stimulation alone did not increase expression above baseline levels (Supplementary Fig. 1E–G). Consistently, inhibition of CD3 signalling at the level of extracellular signal-related kinase (Erk) and Src prevented NRK1 upregulation in CD3/CD28-stimulated CD4+ T cells, as did inhibition of phosphoinositide 3-kinase (PI3K) and Akt within the CD28 signalling pathway. Inhibition of calcineurin also prevented upregulation, indicating a requirement for NFAT activity (Supplementary Fig. 1H). Therefore, NRK1 is upregulated during T cell activation through coordinated activity of CD3 and CD28 signalling pathways.

### NR increases NAD/H levels in stimulated human CD4+ T cells, but regulates activation and effector function

Next, to assess the contribution of NRK1 to NAD/H levels in human CD4+ T cells, these were cultured in the presence of the substrate, NR,

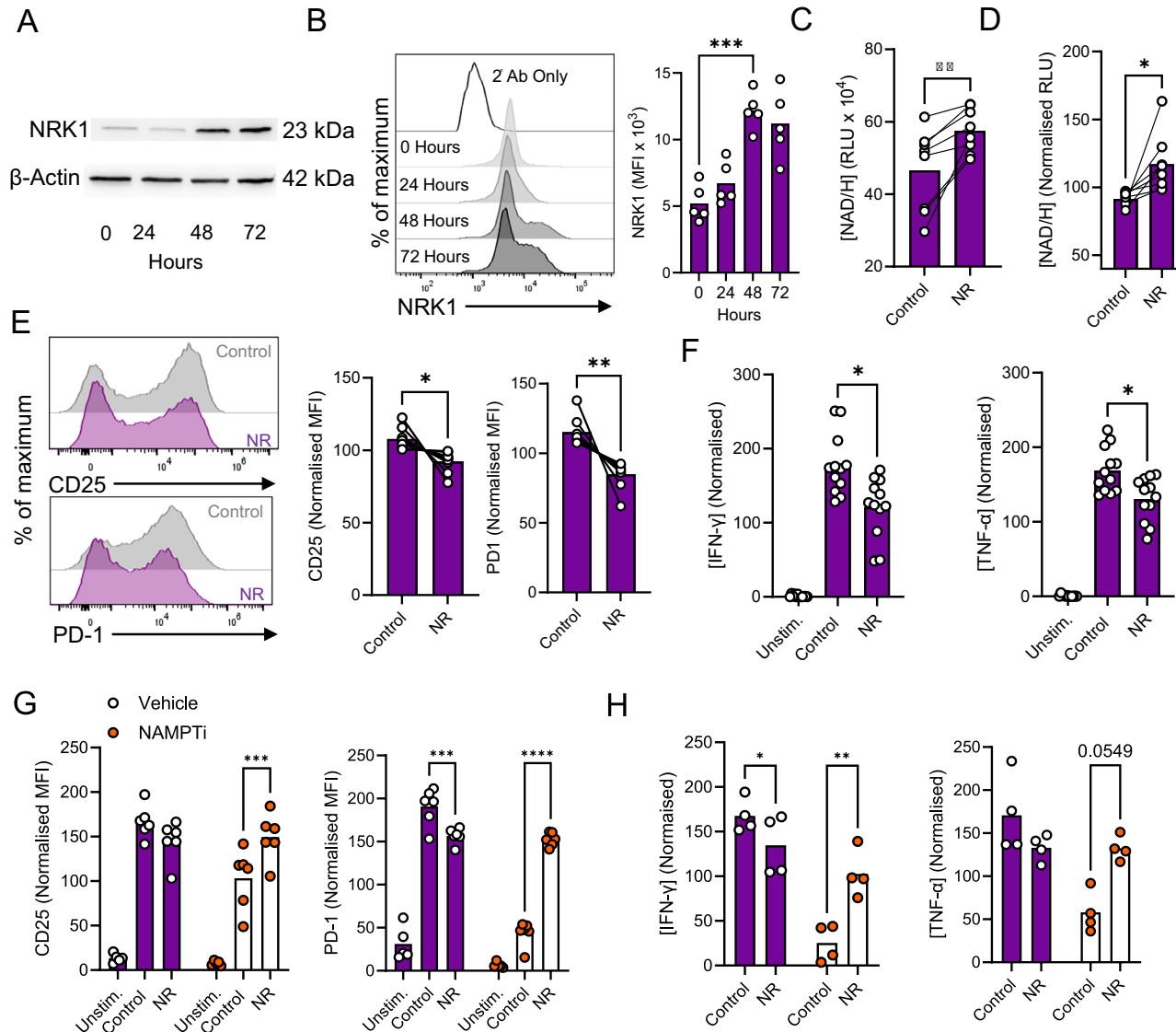

**Fig. 1 | NRK1 is upregulated in stimulated CD4⁺ T cells and NR suppresses their activation and function. A,B** Human CD4⁺ T cells were stimulated via CD3/CD28 for indicated time points and assessed for NRK1 protein abundance by (**A**) western blot and (**B**) flow cytometry (**A**, one representative experiment; **B**, representative histograms and summarised for n = 5 independent donors). **C,D** Human CD4⁺ T cells were stimulated via CD3/CD28 for 48 h ± 0.5 mM NR and assayed for total NAD/H abundance (summarised for n = 7 independent donors, **C** raw data as relative luminescence units (RLU), **D** normalised values). **E–H** Human CD4⁺ T cells were stimulated as in (**C, D**) or additionally in presence of 0.1 μM NAMPTi (FK866) where indicated and assessed for (**E, G**) CD25 and PD-1 expression by flow cytometry (representative histograms and summarised for (**E**) n = 7 independent donors, (**G**) n = 6 independent donors) and (**F, H**) IFN-γ and TNF-α secretion by ELISA (summarised for (**F**) n = 12 independent donors and (**H**) n = 4 independent donors). Where normalised, data are expressed as a percentage of the mean value across all samples analysed for each individual donor. p values were calculated by (**B, F**) two-sided repeated measures ANOVA or (**G, H**) two-way ANOVA with Holm-Sidak's post-hoc test, and (**C–E**) two-sided paired t test *p < 0.05, **p < 0.01. **B** p = 0.0004, **C** p = 0.0044 **D** p = 0.0188 **E** p = 0.0104, p = 0.0015 **F** p = 0.0344, p = 0.0480 **G** p = 0.0010, p = 0.001 p < 0.0001, **H** p = 0.0252, p = 0.0046. Source data are provided as a Source Data file.

and assessed for total intracellular NAD/H abundance (combined NAD+ and NADH). This confirmed NR provision significantly increased NAD/H abundance in activated CD4⁺ T cells (Fig. 1C, D). NR provision did not alter NRK1 expression in activated cells (Supplementary Fig. 1I). NR treatment also rescued NAD/H levels in the presence of the NAMPT inhibitor, FK866 (NAMPTi). In these experiments, failure of NAM to restore NAD/H in the presence of FK866 confirmed effective enzyme inhibition and indicated that NR supplementation increases NAD/H levels via NRK1 (Supplementary Fig. 1J).

Having established that NRK1 significantly contributes to NAD/H synthesis in human CD4⁺ T cells, we then interrogated the implications of this for the immune functionality of these cells. We first assessed activation status upon CD3/CD28 stimulation in the presence of NR,

compared to untreated control cells. We also assessed secretion of the effector cytokines IFN-gamma (IFN-γ) and TNF-α. We somewhat surprisingly observed that provision of NR consistently decreased T cell activation compared to control conditions, as indicated by decreased expression of the activation markers CD25 and PD-1 (Fig. 1E), albeit not the early activation marker CD69 (Supplementary Fig. 1K). Consistent with decreased activation, NR treatment also substantially reduced secretion of IFN-γ and TNF-α (Fig. 1F). Taken together, these observations suggest that NRK1 activity suppresses rather than promotes CD4⁺ T cell function by contributing to NAD/H synthesis. Of note, when NR was provided in the presence of the NAMPT inhibitor FK866, rescue of activation, IFN-γ and TNF-α secretion was observed (Fig. 1G, H), similar to NAD/H levels (Supplementary Fig. 1J). These results indicate that a

threshold of NAD/H abundance is required to support CD4$^+$ T cell activation and function, in agreement with a recent study[11]. NRK1 activity can contribute to reaching this threshold when other contributing synthesis or salvage pathways are limited. However, in the context of unhindered activity of other pathways, our data indicate that NRK1 activity generally restrains CD4$^+$ T cell activity.

## NRK1-deficient murine CD4$^+$ T cells demonstrate hyper-functionality but impaired survival

As yet, specific pharmacological inhibitors of NRK1 are not described. Therefore, to better understand the role of this enzyme in CD4$^+$ T cells, a genetically-modified murine model was employed, where NRK1 was not expressed in any cell type (NRK1KO). Assessment of T cell populations and differentiation status in these mice confirmed thymic and splenic T cell populations at key stages of differentiation to be at similar frequencies in NRK1KO animals compared to wild-type (WT) littermate controls. Additionally, splenic CD4$^+$ T cell populations, which would be used for in vitro experiments, demonstrated similar maturation status (Supplementary Fig. 2A–E). Analysis of NRK1 transcript levels by qPCR identified that, as in human CD4$^+$ T cells, activation increased NRK1 expression in WT splenic murine CD4$^+$ T cells, but, importantly, not in those from NRK1KO animals (Fig. 2A). Further qPCR analyses identified that, again as in human cells, NRK2 transcripts were not detectable. We additionally observed that NAMPT demonstrated expected increases in expression upon murine T cell stimulation, which were comparable in WT and NRK1KO cells (Supplementary Fig. 2F). Of note, two isoforms of NMNAT (NMNAT1 and 3), which convert NRK1-generated NMN into NAD, demonstrated greater transcript abundance in stimulated NRK1KO cells compared to WT (Supplementary Fig. 2G), which may reflect a cellular adaptation to decreased NRK1 activity, particularly upon T cell stimulation. Despite this, measurement of intracellular NAD/H abundance demonstrated lower NAD/H abundance in NRK1KO CD4$^+$ T cells compared to WT, which also did not further increase upon provision of NR, confirming the lack of NRK1 activity (Fig. 2B). Further confirmation of this was provided upon NAMPT inhibition. Here, NAD/H levels decreased substantially, similar to unstimulated cells, and NR could rescue these in WT but not NRK1KO cells (Fig. 2C).

Implications of NRK1 deficiency for CD4$^+$ T cell function were next assessed by stimulating WT vs. NRK1KO T cells within splenocyte cultures and assessing activation and cytokine expression. In these experiments, the activation markers CD69, CD25 and PD-1 demonstrated similar stimulation-induced upregulation in WT and NRK1KO cells (with gating strategy and controls Supplementary Fig. 2H–J). However, NRK1KO cells demonstrated markedly increased capacity to express the effector cytokines IFN-γ, TNF-α and IL-2 upon stimulation (Fig. 2D–G). For IFN-γ and IL-2 this was reflected as a greater frequency of cytokine-expressing cells, whereas for TNF-α, similar frequencies of WT and NRK1KO cells were cytokine-positive, but NRK1KO cells had a higher mean fluorescence intensity (MFI) (Fig. 2D–G, Supplementary Fig. 2K–M). Consistent with observations in human cells, the addition of NR to culture decreased cytokine expression in WT cultures. This did not occur in NRK1KO cells, confirming NRK1-dependency (Fig. 2D–G, Supplementary Fig. 2K–M).

To understand whether observed increases in cytokine expression were CD4$^+$ T cell-intrinsic or related to effects of NRK1 deficiency within other cells in the splenocyte culture, CD4$^+$ T cells were purified, stimulated and assessed for IFN-γ and TNF-α production by ELISA. Similar to flow cytometry of splenocyte cultures, ELISA analysis identified increased capacity of NRK1KO CD4$^+$ T cells to secrete IFN-γ upon stimulation. This was evident at 8 h post-stimulation (Fig. 2H), but differences in total secreted cytokine between WT and NRK1KO cells started to equalise at 24 h of culture and were no longer apparent for either cytokine at 48 h. Since cytokine secreted to the supernatant reflects both the function and the total number of cells present, one

explanation for this could be greater loss of NRK1KO cells during the 48-h experiment. This was therefore directly assessed by quantifying frequencies of viable CD4$^+$ T cells, which confirmed these were decreased in stimulated NRK1KO vs. WT CD4$^+$ T cells at 48 h (Fig. 2I). In unstimulated cells, similar viability was observed between WT and NRK1KO cultures, reflecting that their disparity in NRK1 activity is much greater upon stimulation, when NRK1 is significantly upregulated in WT cells. Therefore, NRK1-deficient murine CD4$^+$ T cells demonstrate hyper-functionality upon activation, consistent with observations in human cells that NR supplementation limits T cell activation and function. NRK1 also appears to importantly contribute to CD4$^+$ T cell survival upon activation, which is decreased upon NRK1 loss.

## NRK1-deficient CD4$^+$ T cells demonstrate modest metabolic alterations, including decreased glycolysis

NAD/H is a critical redox cofactor for central metabolic pathways including glycolysis, the TCA cycle and activity of the mitochondrial ETC. We therefore interrogated how NRK1 activity impacted the metabolic capacity of CD4$^+$ T cells. First, extracellular flux analysis was conducted to measure rates of mitochondrial oxygen consumption and glycolysis. These assays were performed on cells stimulated for 48 h with anti-CD3/CD28, since NRK1 is upregulated upon T cell activation. These experiments identified that NRK1KO cells demonstrated similar basal and ATP-coupled mitochondrial oxygen consumption rates (OCR) as WT cells, but decreased maximal OCR after mitochondrial uncoupling (Fig. 3A–C and Supplementary Fig. 3A). Parallel analysis of glycolysis, as measured by extracellular acidification rate (ECAR) identified a consistent decrease in basal glycolysis, as well as maximal rates of glycolysis measured after ETC inhibition (Fig. 3D–F). When analysed as a ratio, NRK1KO cells consistently demonstrated reduced basal ECAR/OCR than WT cells (Fig. 3G), indicating NRK1 activity is more important for supporting glycolysis than mitochondrial OXPHOS in CD4$^+$ T cells. In agreement, when NR was provided, WT but not NRK1KO cells increased basal glycolysis rates, whilst basal OCR was largely unchanged (Fig. 3H and Supplementary Fig. 3B). Additionally, WT and NRK1KO CD4$^+$ T cells demonstrate similar mitochondrial mass and membrane potential, as assessed by flow cytometry analysis of mitotracker probe fluorescence (Supplementary Fig. 3C, D).

As an orthogonal approach to assess metabolic pathway usage of these cells, the SCENITH assay was employed[20]. This assesses rates of protein synthesis as a surrogate for cellular metabolic status and interrogates how glucose metabolism vs. mitochondrial OXPHOS contribute, using specific inhibitors to infer relative activity of these pathways. This approach identified that overall protein synthesis, as measured by ribosomal puromycin incorporation, was decreased in activated NRK1KO compared to WT (Fig. 3I, J, gating strategy Supplementary Fig. 3E). In addition, probing pathway contributions revealed mitochondrial dependence was consistently greater in NRK1KO than WT cells, with glycolytic capacity reciprocally decreased (Fig. 3I, K, L). Therefore, overall rates of protein synthesis are slightly decreased in the absence of NRK1 activity, and there is a shift towards this being more dependent upon mitochondrial OXPHOS vs. glycolysis, in agreement with the extracellular flux analysis data indicating reduced glycolysis in NRK1KO cells.

Finally, to interrogate metabolic pathway usage in more detail, stable isotope-based metabolic tracing was performed. WT and NRK1KO CD4$^+$ T cells were activated and cultured with either fully labelled $^{13}$C-glucose or $^{13}$C-glutamine (10 mM and 2 mM, respectively) prior to analysis of intracellular metabolites by mass spectrometry. It has been previously observed that in activated T cells, glutamine is a larger contributor of carbons to the TCA cycle than glucose[21,22]. Here, consistent with this, TCA cycle intermediates demonstrated in the range of 0.4 to 0.6 fractional labelling from $^{13}$C-glutamine, compared to ~0.2 from $^{13}$C-glucose (Fig. 3M and Supplementary Fig. 3F, G). When

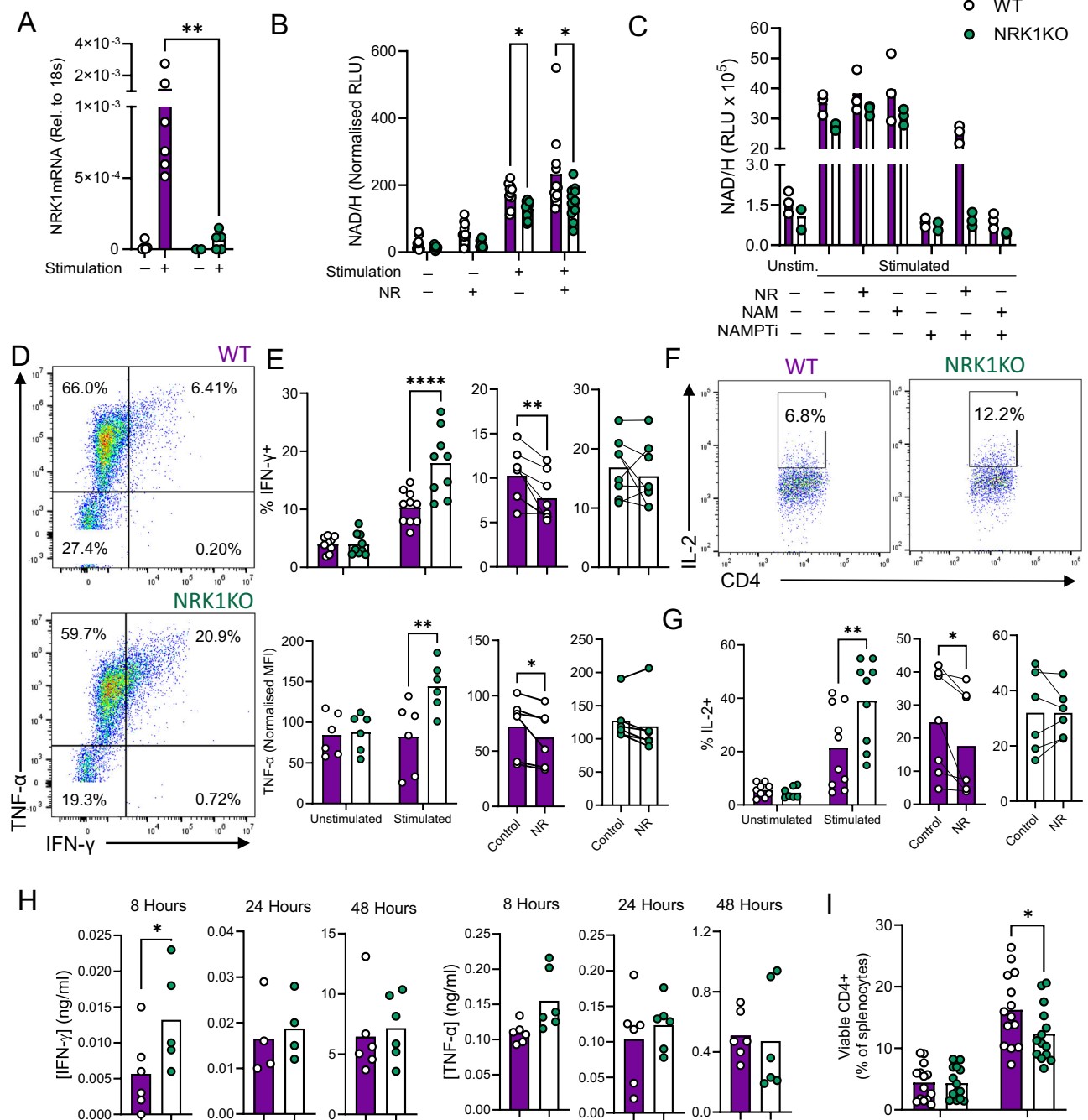

**Fig. 2 | NRK1-deficient CD4+ T cells demonstrate decreased NAD/H abundance, hyper-functionality and impaired viability. A** Murine splenic CD4+ T cells were isolated from littermate WT or NRK1KO animals, cultured ± stimulation via CD3/CD28 for 48 h, and assessed for NRK1 transcript abundance by qPCR (summarised for $n = 5$ individual animals, shown relative to 18 s). **B** Murine CD4+ T cells cultured as in (**A**) ± 0.5 mM NR were assessed for total NAD/H abundance by bioluminescent assay (summarised for $n = 10$ WT and $n = 11$ NRK1KO animals). **C** Stimulated WT or NRK1KO murine CD4+ T cells were treated with indicated compounds (0.5 mM NR, 0.5 mM NAM, 0.1 µM FK866 (NAMPTi)) and assessed for total NAD/H abundance as in (**B**) (summarised for $n = 3$ individual animals). **D,G** CD4+ T cells within murine splenocyte cultures were stimulated via CD3/CD28 for 48 h ± 0.5 mM NR and assessed for abundance of (**D, E**) IFN-γ, TNF-α and (**F, G**) IL-2 by intracellular cytokine staining; representative dot plots for **D** IFN-γ/TNF-α and **F** IL-2; cytokines summarised for: **E** (upper) $n = 10$ WT and $n = 9$ NRK1; (lower) $n = 6$ WT and $n = 6$

NRK1KO, **G** $n = 10$ WT and $n = 9$ NRK1KO animals). **H** Murine CD4+ T cells as in (**A**) were stimulated via CD3/CD28 for indicated time points and secreted IFN-γ and TNF-α measured by ELISA (summarised for $n = 6$ WT and $n = 6$ NRK1KO animals). **I** CD4+ T cells within murine splenocyte cultures were stimulated via CD3/CD28 for 48 h assessed for viability by live/dead probe staining (summarised for $n = 14$ WT and $n = 14$ NRK1KO animals). Where normalised, data are expressed as a percentage of the mean value across all samples analysed together for a batch of mice, of equivalent numbers of WT and NRK1KO. $p$ values were calculated by (**A**–**C E, G** (left hand plots) and **I** two-sided two-way ANOVA, (**E, G** right hand plots) two-sided unpaired t test or **H** repeated measures ANOVA and Holm-Sidak's post-hoc test and *$p < 0.05$, **$p < 0.01$, ****$p < 0.0001$. **A** $p = 0.0069$, **B** $p = 0.034$, $p = 0.044$ **E** (upper) $p < 0.0001$, $p = 0.0087$, (lower) $p = 0.0061$, $p = 0.0468$, **G** $p = 0.0047$, $p = 0.0330$, **H** $p = 0.0375$, **I** $p = 0.0333$. Source data are provided as a Source Data file.

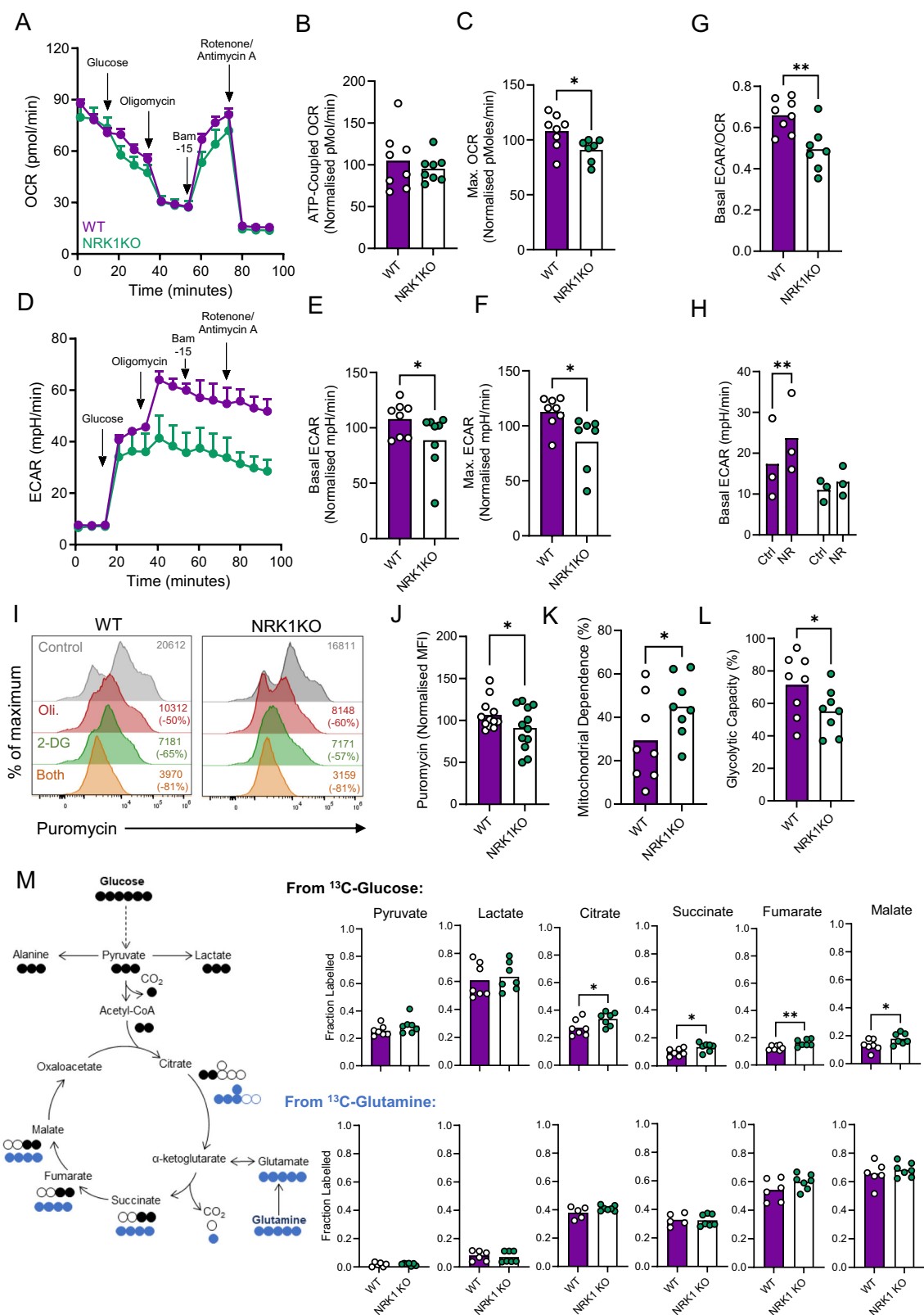

comparing WT vs. NRK1KO cells, consistently increased fractional labelling of TCA cycle intermediates from $^{13}C$-glucose was observed in NRK1KO cells, alongside a trend towards increased pyruvate labelling, but similar labelling of intracellular lactate. Conversely, labelling of TCA cycle intermediates from $^{13}C$-glutamine was comparable between WT and NRK1KO cells, with minimal labelling into pyruvate and lactate, as expected. These $^{13}C$-glutamine tracing data indicate similar

mitochondrial TCA cycle activity per se in NRK1KO and WT cells, consistent with the findings from the other assays. Therefore, increased TCA labelling from glucose may rather reflect increased tendency to oxidise rather than reduce glucose-derived pyruvate in the absence of NRK1 activity. This agrees with the extracellular flux and SCENITH data, which indicate decreased glycolysis in these cells. Overall, however, metabolic changes upon NRK1 loss were relatively

**Fig. 3 | NRK1-deficient CD4⁺ T cells exhibit decreased glycolysis and maximal respiratory capacity. A–H** Splenic murine CD4⁺ T cells were isolated from littermate WT or NRK1KO animals, stimulated via CD3/CD28 for 48 h ± 0.5 mM NR as indicated and assessed by extracellular flux analysis. (**A**) Example oxygen consumption rate (OCR) trace, (**B**) ATP-coupled and (**C**) maximal OCR; (**D**) Example extracellular acidification rate (ECAR) trace (**E**), (**H**) basal and (**F**) maximal ECAR; (**G**) ratio of basal ECAR/OCR (summarised for **B, E** $n = 8$ WT and $n = 8$ NRK1KO, **C, F, G** $n = 8$ WT and $n = 7$ NRK1KO animals) and **H** $n = 3$ WT and $n = 3$ NRK1KO animals). **I–L** Murine CD4⁺ T cells cultured as in (**A–H**) were analysed by flow cytometry for ribosomal puromycin incorporation under control conditions (DMSO) or in presence of Oligomycin (Oli. 1 μM), 2-DG (10 mM) or both; (**I**) representative flow cytometry histograms; (**J–L**) summarised for (**J**) puromycin MFI under control conditions and (**K, L**) mitochondrial dependence and glycolytic capacity calculated as described in the 'methods' section (summarised for $n = 8$ WT and $n = 8$ NRK1KO animals). **M** Murine CD4⁺ T cells isolated as in (**A–H**) were stimulated for 24 h in presence of fully labelled ¹³C-glucose or ¹³C-glutamine and assessed for fractional isotopic labelling of indicated metabolites by GC-MS (summarised for $n = 7$ WT and $n = 7$ NRK1KO animals). Where normalised, data are expressed as a percentage of the average (mean) value across all samples analysed together for a batch of mice, of equivalent numbers of WT and NRK1KO. $p$ values were calculated by (**B, C, E–G, J–M**) unpaired, two-sided t test and **H** two-sided two-way ANOVA and Holm-Sidak's post-hoc test. *$p < 0.05$, **$p < 0.01$. **C** $p = 0.0322$, **E** $p = 0.0476$, **F** $p = 0.0205$, **G** $p = 0.0052$, **H** $p = 0.0012$, **J** $p = 0.0515$, **K** $p = 0.0425$, **L** $p = 0.0361$, **M** $p = 0.0240$, $p = 0.0162$, $p = 0.0050$, $p = 0.0242$. Error bars in (**A, D**) represent SEM. Source data are provided as a Source Data file.

modest and unlikely to underpin the large changes in functionality observed in these cells, particularly since decreased glycolysis is rather reported to be linked to decreased cytokine expression[23]. We therefore probed alternative mechanistic explanations for the hyper-functional phenotype of NRK1-deficient CD4⁺ T cells.

## NRK1 activity preferentially promotes NADP/H synthesis in CD4 + T cells, controlling ROS abundance and signalling

In addition to its role as a redox co-factor and enzyme co-substrate, NAD + is phosphorylated to generate NADP+, via the activity of NAD kinase (NADK) 1 and 2. NADP/H supports pentose phosphate pathway (PPP) activity, nucleotide, cholesterol and lipid synthesis, and also plays a critical role in controlling cellular reactive oxygen species (ROS) abundance, signalling, oxidative damage and associated cell death. Specifically, glutathione peroxidase enzymes inactivate peroxides, coupled to the oxidation of glutathione (GSH) to glutathione disulfide (GSSG). GSSG is converted back to GSH by glutathione reductase, requiring oxidation of NADPH to NADP+. It was therefore possible that the increased cell death observed in NRK1KO CD4⁺ T cells could be related to decreased NADP/H abundance, impaired capacity to regenerate GSH, and increased cellular ROS. Moreover, cellular ROS abundance has been directly linked to T cell activation and cytokine expression via promoting nuclear NFAT translocation[1]. To directly probe a role for NRK1 regulating these processes, cellular NADP/H levels (combined NADP+ and NADPH) were first measured in purified CD4⁺ T cells, which identified these to be significantly decreased in NRK1KO cells compared to WT (Fig. 4A). Indeed, relative to WT cells, NADP/H levels were decreased to a greater extent than NAD/H (Fig. 4B). These experiments also confirmed, as expected, that provision of NR increased intracellular NADP/H in WT, but not NRK1KO cells. Of note, the increase in NADP/H abundance observed in WT cells was greater than that of NAD/H upon NR provision, indicating NRK1 activity causes a preferential increase in NADP/H, in agreement with the greater decrease of NADP/H vs. NAD/H observed in NRK1KO cells (Fig. 4C). Consistent with the decreased abundance of NADP/H in NRK1KO cells, decreased GSH/GSSG ratios were also observed, particularly upon activation (Fig. 4D) and increased cellular ROS, as measured by DCFDA fluorescence (Fig. 4E and Supplementary Fig. 4A). Finally, assessment of NFAT abundance within isolated nuclei identified this to be consistently increased within activated, but not resting NRK1KO CD4⁺ T cells compared to WT controls (Fig. 4F and Supplementary Fig. 4B, C), whilst total cellular NFAT levels were comparable (Supplementary Fig. 4D). Therefore, NRK1 activity appears to critically maintain NADP/H abundance in murine CD4⁺ T cells, with significant implications for cellular ROS homoeostasis and signalling. To further confirm the importance of NRK1 activity for NADP/H maintenance, we interrogated another NADP/H-dependent process, PPP activity. The first, rate-limiting step of the PPP is conversion of glucose-6-phosphate to 6-phosphogluconolactone, by glucose-6-phosphate dehydrogenase (G6PD), which is coupled to NADP+ reduction to NADPH. To probe PPP activity, stimulated WT and NRK1KO CD4⁺ T cells were cultured with 1,2-¹³C-glucose for 24 h and analysed for intracellular metabolite labelling. This identified decreased abundance of ¹³C-labelled isotopomers of the PPP intermediates xyulose-5-phosphate and seduheptolose-7-phosphate in NRK1KO cells, indicating decreased PPP activity, albeit labelled upstream ribulose-5-phosphate abundance (generation of which is also coupled to NADP+ reduction) was similar (Fig. 4G). Further analysis of ¹³C-labeling of TCA cycle intermediates in these samples identified increased abundance of M + 2 isotopomers in NRK1KO vs. WT cells, consistent with earlier observations of increased glucose oxidation (Supplementary Fig. 4E). However, this was not accompanied by increased abundance of M + 1 isotopomers, which arise from PPP intermediates re-entering into glycolysis. These data confirm that PPP activity does not increase to the same extent as glucose oxidation in NRK1KO cells, providing further functional evidence of decreased NADP/H abundance and activity.

Next, to directly probe whether limiting availability of NADPH, which is required for the reduction of GSSG to GSH, is sufficient to recapitulate effects of NRK1 deficiency on CD4⁺ T cell ROS abundance and cytokine secretion, cells were treated with a specific G6PD inhibitor to prevent NADP+ reduction to NADPH. This dose-dependently increased cellular ROS levels (Fig. 5A) and elevated IFN-γ and TNF-α expression (Fig. 5B, C). As a complementary approach to directly test whether elevated ROS abundance drives increased nuclear NFAT translocation and hyper-functionality of NRK1KO CD4⁺ T cells, the antioxidant N-acetylcysteine (NAC) was employed. Inclusion of NAC in culture significantly decreased cytokine expression in NRK1KO CD4⁺ T cells, bringing this down to levels observed in WT cells (Fig. 5D, E). Alongside this, increased activated-induced nuclear NFAT translocation in NRK1KO CD4⁺ T cells was no longer observed in the presence of NAC (Fig. 5F and Supplementary Fig. 5A). Indeed, NAC substantially decreased nuclear NFAT abundance in both WT and NRK1KO cells, whilst not affecting total cellular NFAT levels (Supplementary Fig. 5B), highlighting the ROS-dependency of nuclear NFAT translocation. Taken together, these experiments reveal that significant decreases in cellular NADP/H in NRK1-deficient CD4⁺ T cells are associated with impaired regeneration of GSH and increased ROS abundance and signalling. This appears to importantly underpin the hyper-functionality of NRK1KO CD4⁺ T cells, since ROS augmentation is sufficient to promote this in WT cells, whilst ROS scavenging reverses this phenotype in NRK1 deficiency. Next, to confirm whether this mechanism explained the observed suppressive effects of NR on human CD4⁺ T cell activation (Fig. 1), these parameters were also assessed in that system. Here, provision of NR to activated human CD4⁺ T cells also caused a greater increase in NADP/H vs. NAD/H (Fig. 5G), associated with increased GSH/GSSG, decreased cellular ROS (Fig. 5H, I) and decreased nuclear NFAT (Fig. 5J).

## NRK1 expression is particularly increased within the cytoplasm of activated CD4⁺ T cells; co-ordinated increases in NMNAT1 and NADK1 expression permit locally elevated NAD/H and NADP/H synthesis

Taken together, analysis of human and murine CD4⁺ T cells identified NRK1 activity preferentially promotes NADP/H synthesis from NAD/H

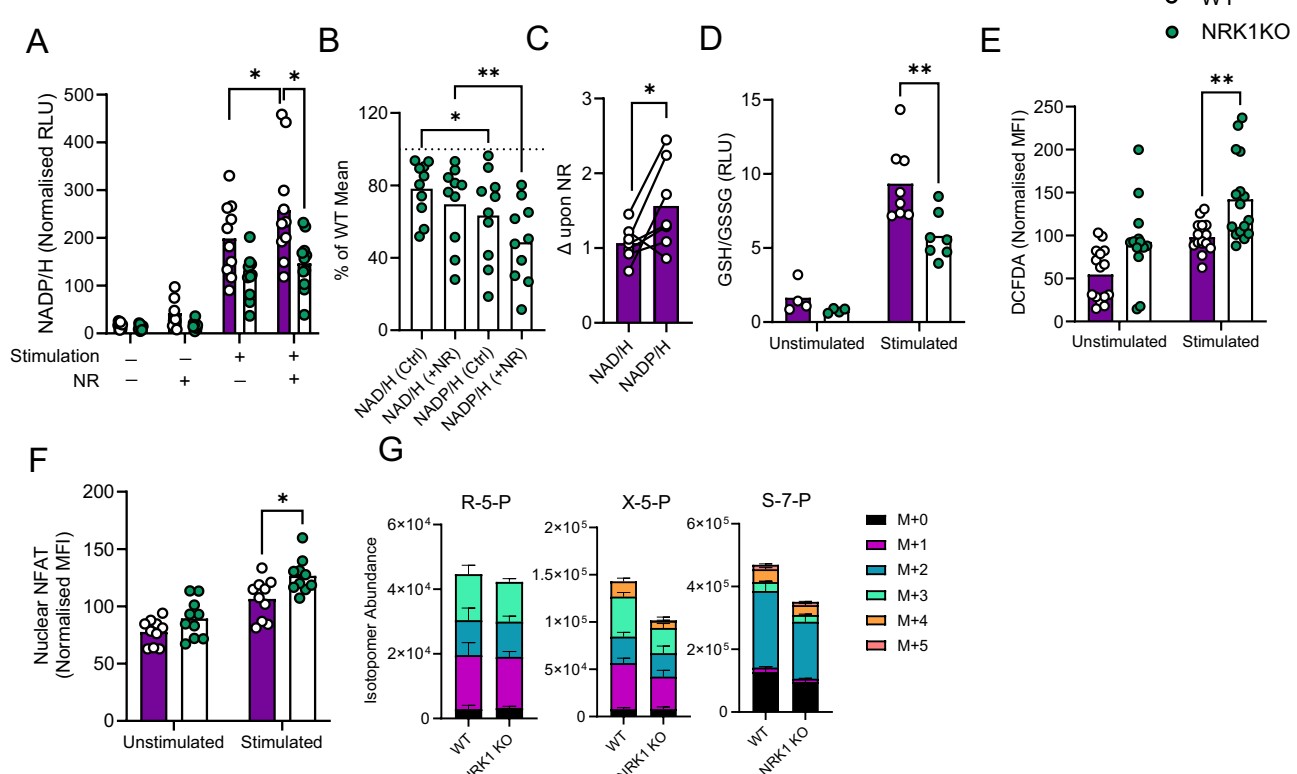

**Fig. 4 | NRK1 activity preferentially promotes NADP/H synthesis in CD4+ T cells.**
**A** Murine CD4+ T cells were isolated from spleens of littermate WT or NRK1KO animals as indicated, cultured ± stimulation via CD3/CD28 and ±0.5 mM NR for 48 h and assessed for total NADP/H abundance by bioluminescent assay (summarised for **A**: $n = 7$ WT and $n = 8$ NRK1KO animals). **B** CD4+ T cell NADP/H measurements for each NRK1KO biological replicate are expressed as a percentage of the mean value of a matched number of WT replicates under the indicated conditions (summarised for $n = 8$ NRK1KO animals). **C** Fold change (Δ) in abundance of NAD/H or NADP/H was calculated for matched murine WT CD4+ T cells, activated as in (**A**), upon provision of 0.5 mM NR during the culture period (summarised for $n = 8$ individual animals per group). **D,F** Murine CD4+ T cells isolated and cultured as in (**A**) were assessed for (**D**) intracellular ratios of GSH/GSSG by bioluminescent assay (summarised for $n = 7$ individual animals per group), (**E**) cellular ROS abundance by

DCFDA fluorescence (summarised for $n = 15$ individual animals per group), (**F**) abundance of NFAT within nuclei, isolated by sucrose gradient centrifugation and assessed by intracellular flow cytometry (summarised for $n = 10$ individual animals per group). **G** Murine CD4+ T cells isolated as in (**A**) were stimulated for 24 h in the presence of 1,2 labelled [13]C-glucose and assessed for fractional isotopic labelling of indicated metabolites by GC-MS (summarised for $n = 4$ individual animals per group). Where normalised, data are expressed as a percentage of the mean value across all samples analysed together for a batch of mice, of equivalent numbers of WT and NRK1KO. $p$ values were calculated by (**A, D, E, F, G**) two-sided two-way or **B** repeated measures ANOVA and Holm-Sidak's post-hoc test, and **C** two-sided paired t test. *$p < 0.05$, **$p < 0.01$,. **A** $p = 0.0389$, $p = 0.0465$, **B** $p = 0.0185$, $p = 0.0021$, **C** $p = 0.0272$, **D** $p = 0.0029$, **E** $p = 0.0047$. **G**: Error bars represent SEM. Source data are provided as a Source Data file.

upon activation, which regulates cellular ROS homoeostasis and nuclear NFAT translocation to restrain CD4+ T cell inflammatory potential and promote survival. It was plausible that this related to subcellular localisation of activity, since cytoplasmic NADP/H synthesis via NADK1 is specifically implicated in ROS homoeostasis[24]. To explore this, NRK1 localisation was first assessed in unstimulated and stimulated human CD4+ T cells by confocal microscopy. This identified that substantial increases in cytoplasmic NRK1 abundance activated cells, with smaller increases observed within nuclei (Fig. 6A). Additionally, combined analysis with heat shock protein 60 (HSP60) as a mitochondrial maker identified limited colocalization, indicating NRK1 does not localise to mitochondria of CD4+ T cells, in agreement with reports in other cell types (Supplementary Fig. 6A). Next, RNA-sequencing data was interrogated for expression of NMNAT isoforms, which convert NRK1-generated NMN into NAD, as well as for expression of NADK isoforms. This revealed increased expression of nuclear/cytoplasmic NMNAT1 in stimulated vs. unstimulated CD4+ T cells, whilst expression of the mitochondrial isoform of this enzyme, NMNAT3, was unchanged. Of note, NMNAT2, which is described to associate with Golgi apparatus/cytoplasm in other cells, was not detected, indicating NMNAT1 generates nuclear and cytoplasmic NAD in CD4+ T cells, and that capacity for this is increased upon activation

(Fig. 6B). Similarly, analysis of this data identified consistent and substantial increases in cytoplasmic NADK1 expression upon activation CD4+ T cells, which was not the case for mitochondrial NADK2 (Fig. 6C). Taken together, these changes in NRK1, NMNAT1 and NADK1 expression and localisation indicate increased capacity for NR-dependent NAD/H and NADP/H synthesis within the cytoplasm of activated CD4+ T cells, which would support the functional effects on ROS homoeostasis and NFAT translocation. Moreover, this would also explain why glycolysis, a cytoplasmic process, but not mitochondrial OXPHOS is impaired within NRK1KO cells.

To directly measure NRK1 activity within subcellular compartments of CD4+ T cells, two complementary techniques were employed. First, genetically-encoded fluorescent NAD biosensors were expressed in primary human CD4+ T cells (Fig. 6D). Specifically, three distinct biosensors were used, which localise to the cytoplasm, nuclei and mitochondria, respectively. Fluorescence of these biosensors is quenched in the presence of NAD, causing a shift in ratios of fluorescence emitted at 530 nm (NAD-sensitive, excitation 488 nm) vs. 510 nm (NAD-insensitive, excitation 405 nm). This is expressed relative to changes in this ratio observed in matched control NAD-insensitive Cp Venus probes[25,26]. Upon NR treatment of activated CD4+ T cells, no quenching of control cpVenus probes was observed (Supplementary

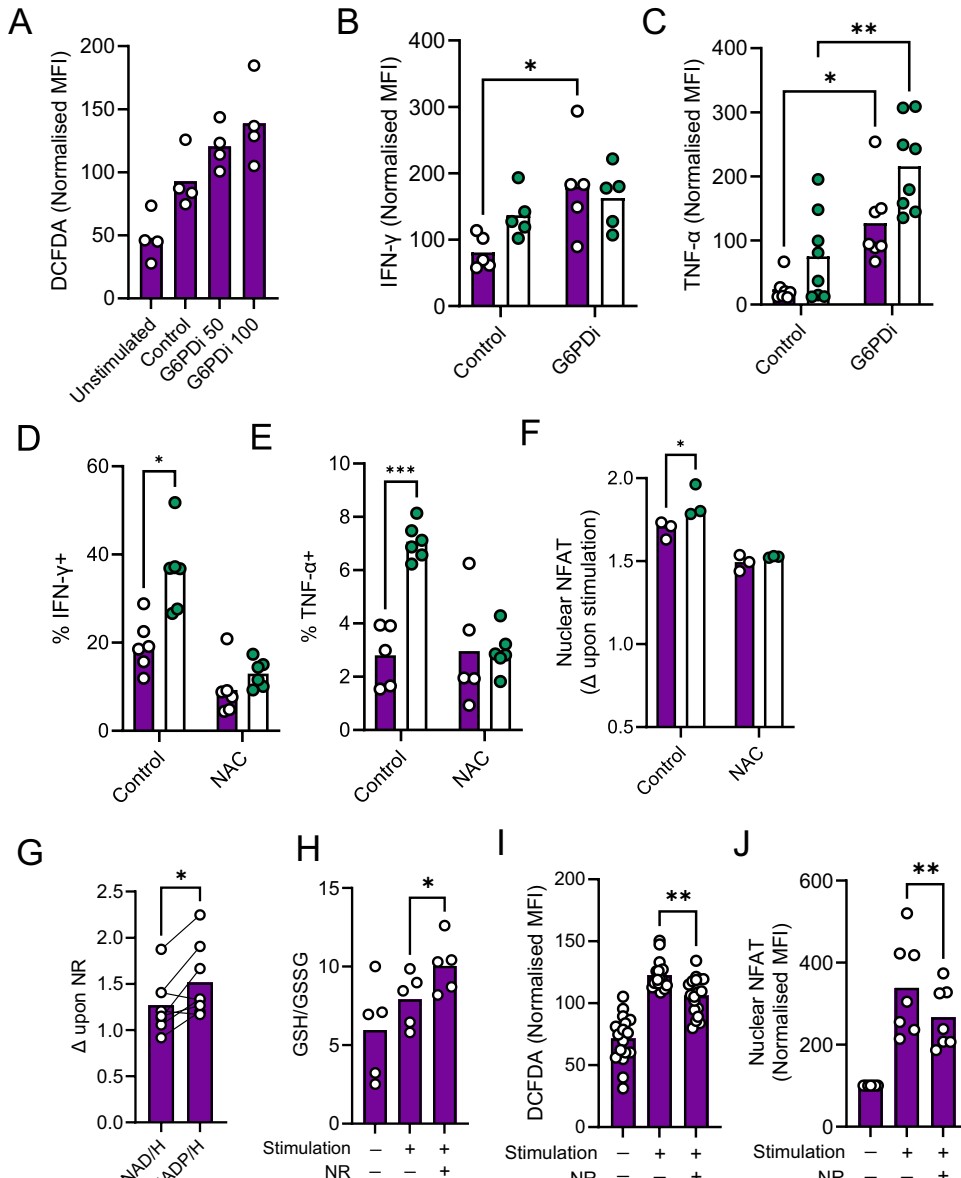

**Fig. 5 | NRK1 activity controls ROS abundance in CD4⁺ T cells, regulating nuclear NFAT translocation. A** Murine CD4⁺ T cells were isolated from spleens of WT mice and stimulated via CD3/CD28 for 48 h ± G6PD inhibitor (G6PDi) at indicated concentrations (nM), then assessed for cellular ROS abundance by DCFDA fluorescence (summarised for $n = 4$ individual animals) **B,C** CD4⁺ T cells isolated from littermate WT or NRK1KO animals as indicated and cultured as in (**A**) were assessed for intracellular abundance of (**B**) IFN-γ and (**C**) TNF-α by flow cytometry (summarised for **B**: $n = 5$ WT and $n = 5$ NRK1KO, **C**: $n = 7$ WT and $n = 8$ NRK1KO animals). Murine CD4⁺ T cells isolated and cultured as in (**A**), additionally in presence of N-acetylcysteine (NAC, 5 mM) were assessed for (**D, E**) intracellular abundance of (**D**) IFN-γ (summarised for $n = 6$ individual animals per group) and (**E**) TNF-α (summarised for B: $n = 5$ WT and $n = 6$ NRK1KO animals) by flow cytometry and (**F**) abundance of NFAT within nuclei (summarised for $n = 3$ individual animals per group). **G–J** Human CD4⁺ T cells were isolated from peripheral blood and

stimulated via CD3/CD28 for 48 h ± 0.5 mM NR and assessed for (**G**) fold change (Δ) in abundance of NAD/H or NADP/H upon provision of 0.5 mM NR (summarised for $n = 8$ independent donors), (**H**) intracellular ratios of GSH/GSSG by bioluminescent assay (summarised for $n = 5$ independent donors), (**I**) cellular ROS abundance (summarised for $n = 18$ independent donors) and (**J**) nuclear NFAT abundance (summarised for $n = 6$ independent donors). Where normalised, data are expressed as a percentage of the mean value across all samples analysed together for a batch of mice, of equivalent numbers of WT and NRK1KO or, for human samples, as a percentage of the mean value across all samples analysed for each individual donor. $p$ values were calculated by **B–F** two-sided two-way or **H–J** repeated measures ANOVA and Holm-Sidak's post-hoc test, and **G** two-sided paired t test. *$p < 0.05$, **$p < 0.01$, ***$p < 0.001$, ****$p < 0.0001$. **B** $p = 0.0139$, **C** $p = 0.0116$, $P = 0.0017$ **D** $p = 0.0508$, **E** $p = 0.0047$, **F** $p = 0.0268$ **G** $p = 0.0194$, **H** $p = 0.0289$, **I** $p = 0.0018$, **J** $p = 0.0042$. Source data are provided as a Source Data file.

Fig. 6B), but quenching of the cytoplasmic and nuclear NAD biosensor probes was observed. Of note, no mitochondrial biosensor probe quenching occurred, consistent with observed NRK1 localisation (Fig. 6E, F). Again, NR, but not NAM, reversed NAMPTi-induced increases in biosensor fluorescence (i.e. decreased NAD, Fig. 6F), confirming NRK1 specificity. Use of these probes therefore confirmed cytoplasmic NRK1 activity in activated human CD4⁺ T cells.

To complement these findings by an orthogonal approach, NAD/H and NADP/H abundance were quantified within cytoplasmic fractions of CD4⁺ T cells recovered after digitonin treatment. This treatment effectively permeabilised the plasma membrane, but did not permeabilise the mitochondrial membrane, as indicated by presence of the cytoplasmic signalling protein ZAP70, but not mitochondrial COXIV within recovered cytoplasmic fractions (Supplementary

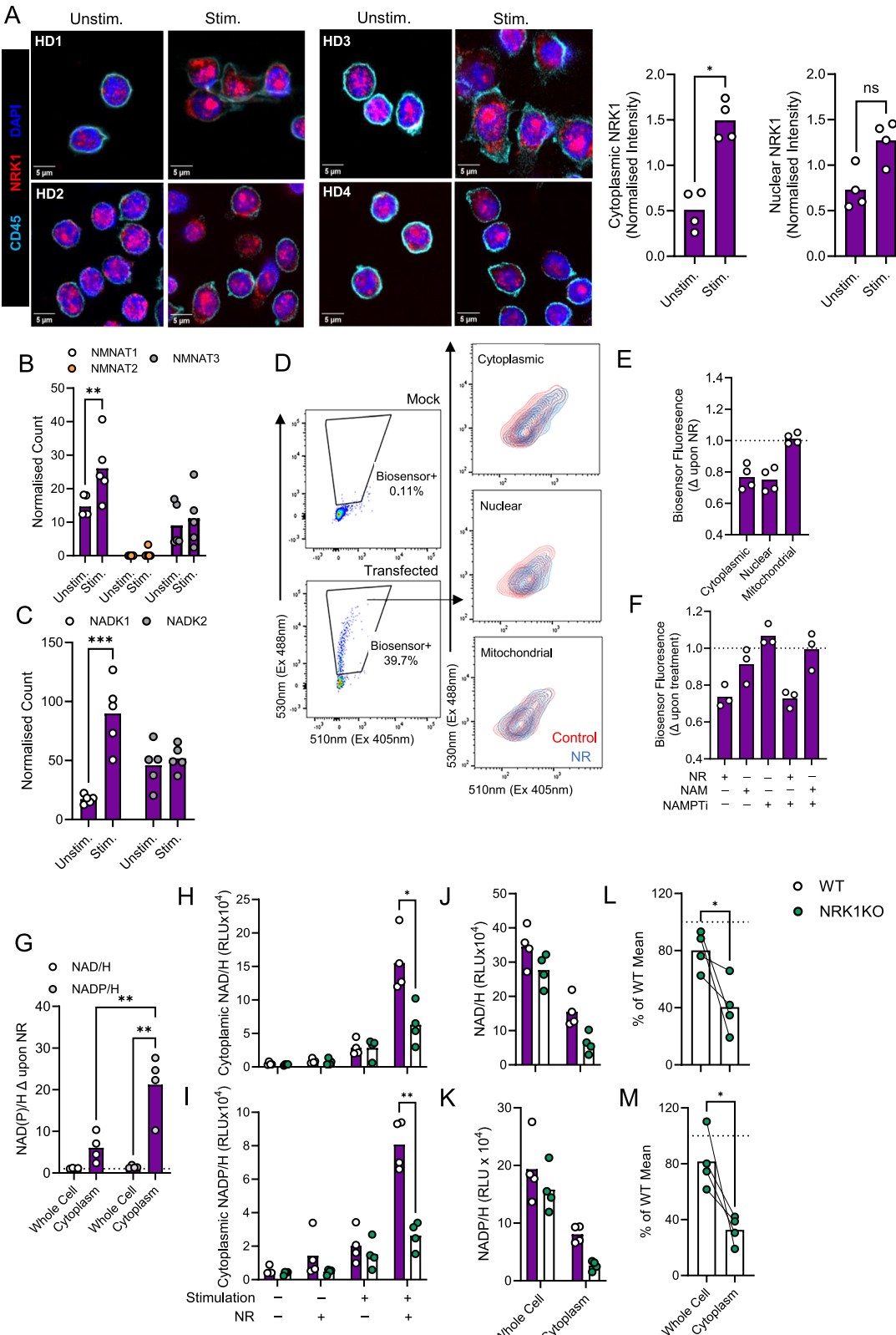

Fig. 6C). Analysis of NAD/H and NADP/H abundance within cytoplasmic fractions identified these were substantially increased by prior NR provision to human CD4$^+$ T cells, indeed to a greater extent than observed in matched intact cells that were analysed in parallel without digitonin treatment (Supplementary Fig. 6D and Fig. 6G). These assays therefore confirm locally elevated NRK1 activity within the cytoplasmic compartment. Moreover, as observed in whole cells, NADP/H was also

increased to a greater extent than NAD/H by NR within cytoplasmic fractions, confirming efficient NADK1 activity, consistent with its increased expression. Application of this analysis to murine WT and NRK1KO CD4$^+$ T cells confirmed substantial increases in cytoplasmic abundance of NAD/H and NADP/H upon activation and NR provision, which was NRK1-dependent (Fig. 6H, I). Moreover, NAD/H and NADP/H were decreased to a much greater extent in cytoplasmic fractions of

**Fig. 6 | NRK1 expression increases in the cytoplasm of activated CD4⁺ T cells, where it locally promotes NAD/H and NADP/H synthesis. A** Confocal microscopy of human CD4⁺ T cells cultured ± CD3/CD28 stimulation for 48 h, for NRK1, DAPI (nuclear marker), CD45 (cell membrane); $n = 4$ independent donors (HD). Cytoplasmic and nuclear NRK1 density, defined using a nuclear mask, expressed for all cells within image, relative to the mean of both conditions for each donor. **B, C** Normalised transcript counts for indicated genes from RNA-sequencing of naïve human CD4⁺ T cells, cultured as in (**A**) ($n = 6$ independent donors). **D,F** Analysis of NAD biosensors in activated human CD4⁺ T cells treated with indicated compounds for 8 h (**D**) representative flow cytometry plots of mock- and biosensor-transfected cells and summarised data for (**E**) cytoplasmic, nuclear and mitochondrial biosensors and (**F**) cytoplasmic biosensor ($n = 3$ independent donors, data expressed as ratio of biosensor fluorescence at 530 nm/510 nm, relative to this ratio in control CpVenus-transfected cells and vehicle-treated cells). **G** Human CD4⁺ T cells as (**A**) and (**H–M**) murine CD4⁺ T cells of WT and NRK1KO

mice, stimulated via CD3/CD28 for 48 h ± 0.5 mM NR, were left intact or digitonin-treated to permeabilise the plasma membrane and assessed for NAD(P)H abundance within whole cells or cytoplasmic fractions. Data are expressed as (**G**) fold change (Δ) in NAD(P)H with NR raw relative luminescence units (RLU) for (**H, I**) cytoplasmic fractions and (**J, K**) whole cell or cytoplasmic fractions from activated cells cultured +0.5 mM NR and whole cell or cytoplasmic (**L**) NAD/H and (**M**) NADP/H from activated cells cultured +0.5 mM NR as a percentage of the mean value of a matched number of WT replicates under the same conditions (**G** $n = 4$ independent donors; **H–M** $n = 4$ individual animals per group) $p$ values were calculated by **A, L, M** two-sided paired t test, **B, C, G, H–K** two-sided two-way ANOVA and **E, F** repeated measures ANOVA and Holm-Sidak's post-hoc test. *$p < 0.05$, **$p < 0.01$. **A** $p = 0.0205$, **B** $p = 0.0019$, **C** $p = 0.0007$, **G** $p = 0.0038$, $p = 0.0018$, **H** $p = 0.0179$, **I** $p = 0.0069$, **L** $p = 0.0345$, **M** $p = 0.0225$. Source data are provided as a Source Data file.

---

activated, NR-treated NRK1KO vs. WT samples than observed for matched intact cells, further confirming this is a key cellular compartment for NRK1 activity in CD4⁺ T cells (Fig. 6J–M). Therefore, cytoplasmic upregulation of NRK1 upon activation of CD4⁺ T cells, alongside concerted increases in NMNAT1 and NADK1 expression, promotes cytoplasmic generation of NAD/H and NADP/H, which supports glycolysis and regulates cellular ROS abundance to promote viability and control NFAT-dependent cytokine expression.

## T cell-intrinsic NRK1 activity maintains effector T cell frequencies and controls pathogen burden during invasive fungal infection

To understand the importance of CD4⁺ T cell-intrinsic NRK1 during an immune response to infection, T cell-specific NRK1KO mice (NRK1Flox+/CD4Cre+) were first generated (Supplementary Fig. 7A–D). Again, phenotyping of thymic and splenic T cell populations confirmed similar T cell differentiation and maturation as in littermate Cre-negative (NRK1Flox+/CD4Cre−) controls (Supplementary Fig. 7E–I), whilst in vitro analyses identified decreased NRK1 transcript abundance in activated, NRK1Flox+/CD4Cre+CD4⁺ T cells (Fig. 7A), but not CD19⁺ B cells, confirming cell type specificity (Supplementary Fig. 7J). Lack of NRK1 activity was also observed in CD4⁺ T cells (Fig. 7B) and, similar to NRK1KO CD4⁺ T cells, increased cytokine expression and impaired viability of NRK1Flox+/CD4Cre+ cells vs. NRK1Flox+/CD4Cre- controls upon in vitro activation (Fig. 7C, D). We also took advantage of this T cell-specific NRK1-deficient system to assess the capacity for CD4⁺ T cells to differentiate into inflammatory subsets in vitro in splenocyte cultures, without confounding effects of NRK1 loss in other splenocytes. Here we observed that under Th1 polarising conditions, NRK1-deficient CD4⁺ T cells demonstrated a greater tendency to differentiate into IFN-γ-expressing cells (gating strategy Supplementary Fig. 8A and Fig. 7E–G). This increased pro-inflammatory potential was again accompanied by reduced viability of NRK1-deficient vs. NRK1-replete cells over this longer period (Fig. 7H). Similarly, under Th17 polarising conditions, increased frequency of NRK1-deficient cells polarised into IL-17-expressing cells, which also co-expressed the Th17-associated transcription factor RORγt (Supplementary Fig. 8B, C). Viability in these cultures was more variable and not significantly different between NRK1-replete and deficient cells (Supplementary Fig. 8D). Conversely, under Th2 polarising conditions, NRK1-deficient cells demonstrated similar capacity as control cells to differentiate into cells expressing the hallmark transcription factor GATA3, in the absence of IFN-γ, and similar viability (Supplementary Fig. 8E–G). Having confirmed the lack of NRK1 expression and activity, and similar functional changes within CD4⁺ T cells in this model compared to those from global NRK1KO mice, infection control and CD4⁺ T cell responses during infection were next studied. These were first assessed during cryptococcosis, an invasive

fungal infection of the lung caused by *Cryptococcus neoformans*, which disseminates to the brain to cause fungal meningitis. It was recently named by the World Health Organisation as the top priority fungal pathogen, requiring new insights into pathogenesis[27]. CD4⁺ T cells are critical for protection, and patients lacking these populations are highly susceptible to disease. Moreover, patient survival and clinical outcomes typically correlate with CD4⁺ T cell frequencies and particularly IFN-γ levels in the cerebrospinal fluid[28,30]. We therefore examined the role of NRK1-dependent control of CD4⁺ T cell activity and fungal burden during this infection. Mice were intranasally administered *C. neoformans* and assessed two weeks later for effector CD4⁺ T cell frequency within the lung and brain, uninvolved inguinal lymph nodes and spleen, as well as lung and brain fungal burden. Comparison of NRK1Flox + /CD4Cre+ mice with NRK1Flox+/CD4Cre- littermate controls, and Cre-recombinase only controls (NRK1Flox−/CD4Cre+) identified consistently decreased frequencies of effector IFN-γ⁺CD4⁺ T cells within brains of NRK1Flox +/CD4Cre+ mice compared to control groups, observed to a lesser extent in the lung (gating strategy Supplementary Fig. 9A and Fig. 8A), and not in the unaffected inguinal lymph node or in the spleen (Supplementary Fig. 9B), highlighting a more important role for NRK1 in maintaining T cell survival upon antigenic stimulation within infected organs compared to uninvolved secondary lymphoid tissues. This decrease in cell frequency may be associated with increased ROS abundance and cell death, as observed in vitro, therefore, γH2AX abundance, as a marker of DNA damage that can be ROS-induced was assessed. Significantly elevated γH2AX was detected in CD4⁺ T cells from NRK1Flox+/CD4Cre+ mice, again particularly within infected organs (Fig. 8B and Supplementary Fig. 9C, D). Increased DNA damage may also result from decreased PARP-mediated repair, which requires NAD as PARP co-substrate, however analysis of PARP activity by measurement of protein poly (ADP- ribose) (PAR) abundance indicated this was similar in activated WT and NRK1KO CD4⁺ T cells (Supplementary Fig. 10A, B). Moreover, measurement of γH2AX abundance within WT and NRK1KO CD4⁺ T cells identified similar levels in cells activated in vitro. This increased upon G6PDi inhibition, confirming NADP/H redox and ROS influence this, and notably increased further in NRK1KO CD4⁺ T cells, where this pathway is already limited by NADP/H availability (Supplementary Fig. 10C). Consistent with these decreases in effector CD4⁺ T cell populations, brain fungal burdens were found to be significantly higher in NRK1Flox+/CD4Cre+ mice than NRK1Flox+/CD4Cre- litter mate controls (Fig. 8C), whilst fungal burdens in the lung were similar (Fig. 8D) Therefore, T cell-intrinsic NRK1 expression is required for an optimal immune response to, and control of this pathogen in the brain, by controlling ROS-induced DNA damage and promoting cell survival. Of note, brain fungal burdens tightly correlated with IFN-γ⁺CD4⁺ T cell frequencies in NRK1Flox+/CD4Cre− mice, but not NRK1Flox+/CD4Cre+ mice,

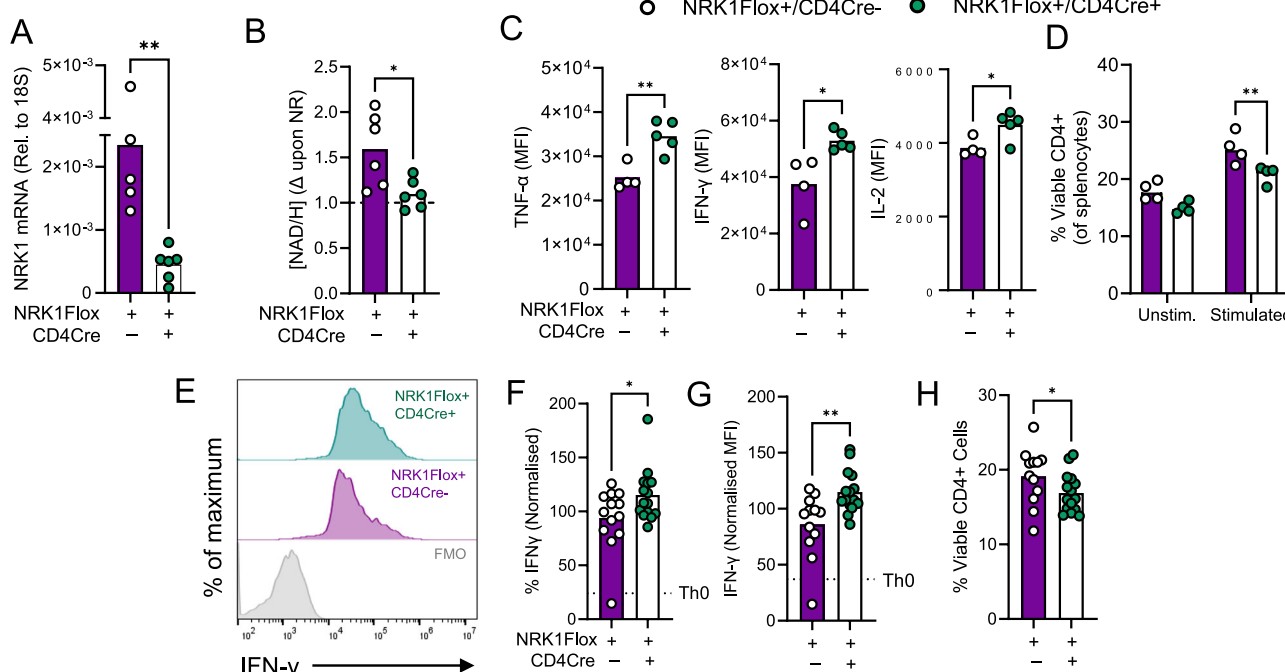

**Fig. 7 | NRK1 activity regulates CD4+ T cell inflammatory function and differentiation in a T cell-intrinsic manner. A–C** Murine CD4+ T cells were isolated from spleens of T cell-specific NRK1KO mice (NRK1Flox+/CD4Cre+) or littermate Cre-recombinase negative controls (NRK1Flox+/CD4Cre−) as indicated, stimulated via CD3/CD28 for 48 h ± indicated compounds and assessed for (**A**) NRK1 transcript abundance by qPCR (summarised for $n = 5$ NRK1Flox+/CD4Cre− and $n = 6$ NRK1Flox+/CD4Cre+ animals, shown relative to 18 s) (**B**) total NAD/H abundance by bioluminescent assay (expressed as fold change (Δ) in NAD/H upon provision of NR during the culture period, summarised for $n = 6$ individual animals per group), (**C**) IFN-γ, TNF-α and IL-2 expression by intracellular cytokine staining (summarised for $n = 5$ individual animals) and (**D**) viability by live/dead probe staining and flow cytometry (summarised for $n = 4$ individual animals). **E–H** Murine CD4+ T cells, isolated from spleens of T cell-specific NRK1KO mice (NRK1Flox+/CD4Cre+) or littermate Cre-recombinase negative controls (NRK1Flox+/CD4Cre−) as in (**A**) were cultured for 6 days under Th1-polarising conditions and assessed for (**E–G**) intra-cellular IFN-γ abundance and (**H**) viability by flow cytometry (summarised for $n = 12$ individual animals per group). Where normalised, data are expressed as a percentage of the mean value across all samples analysed together for a batch of mice, of equivalent numbers of NRK1Flox+/CD4Cre+, NRK1Flox+/CD4Cre− and Cre-recombinase control mice. $p$ values were calculated by **A, C, F–H** two-sided repeated measures or **D** two-way ANOVA and Holm-Sidak's post-hoc test. *$p < 0.05$, **$p < 0.01$. **A** $p = 0.0071$, **C** $p = 0.0037$, $p = 0.0143$, $p = 0.0267$, **D** $p = 0.0091$, **F** $p = 0.0432$, **G** $p = 0.0040$, **H** $p = 0.0416$, Source data are provided as a Source Data file.

further highlighting that NRK1 activity critically supports and maintains infection-elicited effector CD4+ T cell responses (Fig. 8E) in this tissue.

To probe importance of T cell-specific NRK1 in distinct infection context, bone marrow chimaera mice were generated using NRK1Flox+/CD4Cre+ or NRK1Flox+/CD4Cre- bone marrow, and intranasally infected with *Influenza A* virus. After 9 days of infection, mice were assessed for effector CD4+ T cell frequency within the lung and draining mediastinal lymph node (medLN). Mice were also weighed and assessed for overall disease score daily. Comparison of NRK1Flox+/CD4Cre+ chimaera mice with NRK1Flox+/CD4Cre- controls identified significantly fewer viable, donor-derived CD4+ T cells in the lung and medLN at day 9. However, this was also observed in peripheral blood samples taken before infection, indicating poorer engraftment of NRK1-deficient vs. -sufficient CD4+ T cells, potentially related to a role for NRK1 during lymphopenia-induced CD4+ T cell proliferation (Supplementary Fig. 10D, E). To control for this, ratios of donor/host effector CD4+ T cells were assessed within individual tissues, which identified consistently fewer donor CD4+ T cells expressing IFN-γ, IL-2 and TNF-α in medLN of NRK1Flox+/CD4Cre+ chimaeric mice compared to NRK1Flox+/CD4Cre- chimaera controls (Supplementary Fig. 10F and Fig. 8F), although this was not observed in the lung (Supplementary Fig. 10G), despite consistently elevated DNA damage in CD4+ T cells in this tissue (Fig. 8G). Assessment of mouse weight confirmed expected decreases during infection, which were slightly greater in NRK1Flox+/CD4Cre+ chimaeras, albeit variable, particularly in the NRK1Flox+/CD4Cre- group (Fig. 8H). Finally, assessment of

disease score identified that this was consistently greater in NRK1Flox+/CD4Cre+ chimaeras, particularly at day 8 of infection (Fig. 8I), consistent with reduced effector cells observed in the draining medLN.

## Discussion

In this study, how NRK1 activity determines CD4+ T cell biology was interrogated by combined analysis of human primary CD4+ T cell function upon NR provision and assessment of murine NRK1-deficient CD4+ T cells in vitro and in vivo. Key findings include that NRK1 is upregulated upon CD4+ T activation via TCR and CD28 signalling, and that it non-redundantly contributes to cellular NAD/H and NADP/H abundance in this context. Promoting NRK1 activity by the provision of NR suppresses CD4+ T cell activation and effector function. Conversely, NRK1-deficient CD4+ T cells demonstrate a hyper-inflammatory phenotype, robustly expressing effector cytokines but demonstrating impaired viability. Mechanistically, NRK1 activity controls ROS homoeostasis in activated CD4+ T cells, thereby regulating nuclear NFAT translocation. This is associated with preferential support of NADP/H vs. NAD/H synthesis, linked to increased cytoplasmic expression and activity of NRK1 in activated CD4+ T cells, in concert with upregulation of NMNAT1 and NADK1. Indeed, cytoplasmic compartments of NRK1-deficient CD4+ T cells demonstrate substantially depleted NAD/H and NADP/H, and consistently, these cells also exhibit impaired glycolysis but intact mitochondrial OXPHOS. Finally, the importance of CD4+ T cell NRK1 expression was probed in vivo during fungal and viral infections (*Cryptococcus neoformans* and *Influenza A*, respectively), which identified decreased frequencies of effector,

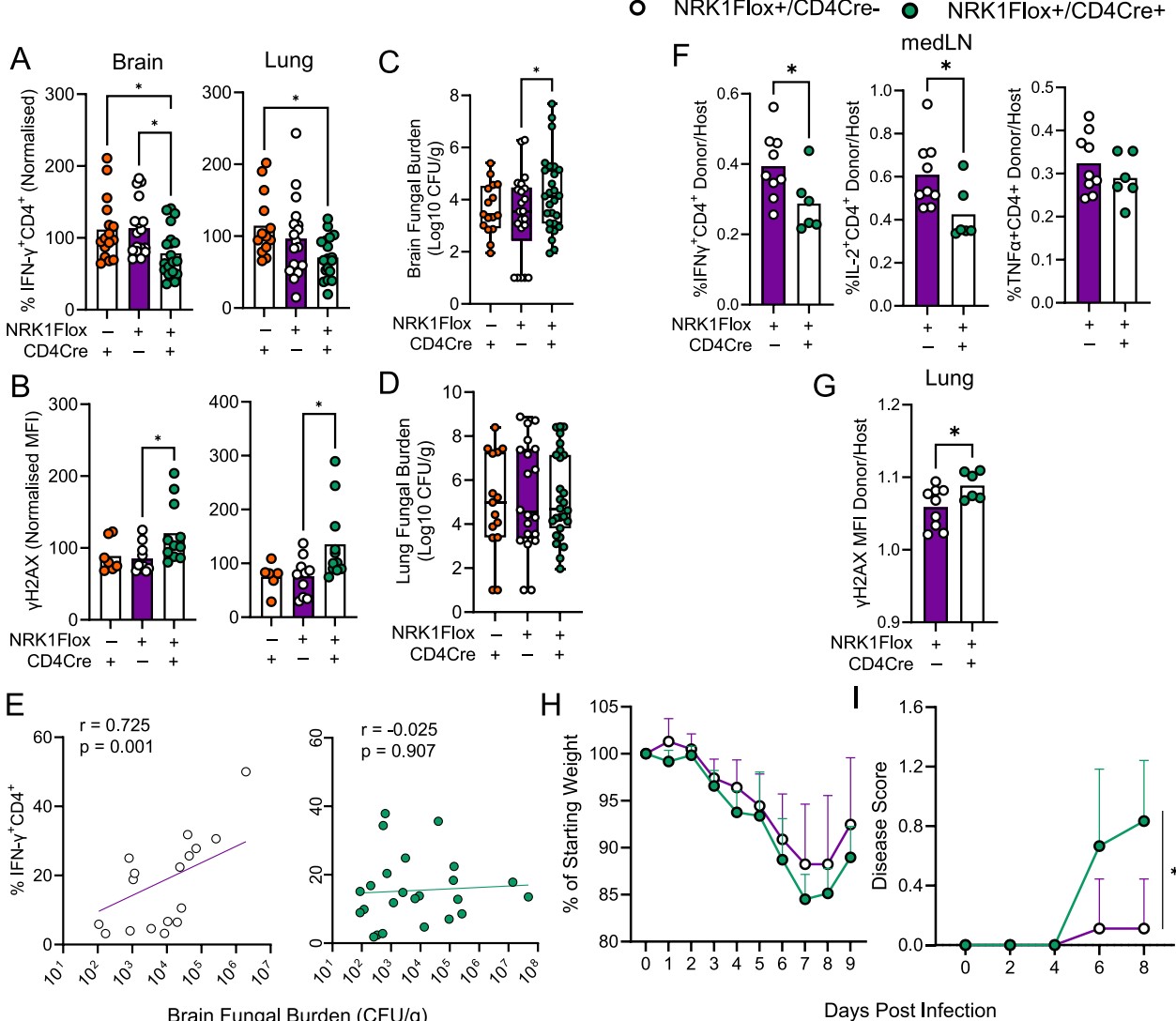

**Fig. 8 | NRK1 activity is required to maintain effector CD4⁺ T cell frequencies and control pathogen burden in vivo during infection. A,E** Cre-recombinase control mice (NRK1Flox-/CD4Cre + ), NRK1Flox + /CD4Cre+ or NRK1Flox + /CD4Cre-mice were infected intranasally with *C. neoformans* and indicated tissues analysed at day 14 for (**A**) frequency of IFN-γ⁺ CD4⁺ T cells and (**B**) DNA damage within CD4⁺ T cells (γH2AX abundance) by flow cytometry; (**C**) brain and (**D**) lung fungal burden and (**E**) correlation between brain fungal burden and frequency of IFN-γ⁺ CD4⁺ T cells, shown with illustrative line of non-linear fit and Spearman correlation data (summarised for **A, E:** n = 16 NRK1Flox-/CD4Cre+, n = 17 NRK1Flox +/CD4Cre− and n = 20 NRK1Flox+/CD4Cre+; **B:** n = 7 NRK1Flox-/CD4Cre+, n = 10 NRK1Flox+/CD4Cre− and n = 11 NRK1Flox+/CD4Cre+; **C, D:** n = 16 NRK1Flox-/CD4Cre+, n = 22 NRK1Flox+/CD4Cre− and n = 27 NRK1Flox+/CD4Cre+ animals), data are compiled from 5 independent experiments). **F−I** Murine bone marrow chimaera recipients of NRK1Flox+/CD4Cre− or NRK1Flox+/CD4Cre+ haemato-poietic cells as indicated were infected intranasally with *Influenza A* and indicated tissues indicated at day 9 for (**F**) frequency of cytokine-expressing CD4⁺ T cells and (**G**) DNA damage within CD4⁺ T cells (γH2AX abundance) by flow cytometry (expressed as a ratio within donor (CD45.2)/host (CD45.1) cells). **H, I** Mice infected as in (**F, G**) were assessed daily during infection for (**H**) weight loss and (**I**) disease score. (**F−I**, n = 9 NRK1Flox+/CD4Cre−, n = 6 NRK1Flox+/CD4Cre+ animals). Where normalised, data are expressed as a percentage of the mean value across all samples analysed together for a batch of mice, of equivalent numbers of NRK1Flox +/CD4Cre+, NRK1Flox+/CD4Cre− and Cre− recombinase control mice. For box plots, line is at median, box extends from 25th to 75th percentile and whiskers are min to max. **H, I** Error bars show SD. p values were calculated by **A−D** two-sided repeated measures ANOVA and Holm-Sidak's post-hoc test **F, G** two-sided unpaired t test, **E** spearman correlation and (**H, I**) two-sided two-way ANOVA and Holm-Sidak's post-hoc test. *p < 0.05, **p < 0.01. **A** (Brain) p = 0.0254, p = 0.0267 (Lung p = 0.0212 **B** (Brain) p = 0.0469 (Lung) p = 0.0430, **C** p = 0.0392, **F** p = 0.0444, p = 0.0352 **G** p = 0.0392 **I** p = 0.0024. Source data are provided as a Source Data file.

cytokine-expressing CD4⁺ T cells within infected tissues (*Cryptococcus*) and draining lymph nodes (*Influenza*) upon T cell-specific NRK1-deficiency, alongside substantially elevated DNA damage. In cryptococcal infection, this was associated with reduced fungal control in the brain, but not the initial site of infection, the lung, highlighting potential different roles of CD4⁺ T cells for pathogen control in these distinct tissue sites. During influenza infection, mice with T cell-specific NRK1 deficiency exhibited elevated disease scores. Therefore, NRK1 activity critically supports CD4⁺ T cell survival and function during infection, with implications for disease outcome. The observation that NRK1

activity plays a largely pro-survival and immunosuppressive role in CD4⁺ T cells agrees with accumulating studies indicating immunoregulatory potential for NR, when orally administered or provided in vitro[5,6,16,18]. Moreover, it is consistent with a report that NR administration augments tissue frequencies of Th1 and Th2 CD4⁺ T cells in a murine sepsis model[17].

Our data additionally identify that the immunosuppressive effect of NR is dependent upon cellular NAD/H status, with NR not inhibiting but rather rescuing cytokine expression under NAMPT inhibition, associated with restoration of NAD/H levels. Similarly, a previous study

reported that augmenting cellular NAD/H abundance by inhibiting NAD-consuming CD38, PARP and SARM1 activity also decreased cellular ROS levels and suppressed inflammatory IL-17 release[18]. Taken together, these data identify cellular NAD/H status as a rheostat for CD4[+] T cell inflammatory activity, likely via balancing metabolic capacity and ROS abundance. However, since NAMPT and NRK1 are coordinately upregulated during T cell activation, it is unlikely NRK1 ever compensates for limited NAMPT under physiological scenarios, but rather more probable it plays an immunoregulatory role, as indicated by NR supplementation studies[5,16,18]. Of note, a recent study reported NAM supplementation similarly suppressed T cell CD25 upregulation upon activation, as well as effector cytokine expression in vitro and in vivo, providing further support for the rheostat concept in the context of other NAD precursors. In this study, NAM supplementation also increased T cell glycolysis, but not mitochondrial OXPHOS, suggesting it may, similarly to NR, particularly augment NAD/H levels within the cytoplasm. Given this was analysed in activated T cells, increased cytoplasmic NMNAT1, as we identify here, may mediate this, albeit the effects of NAM on mTOR activity were also proposed to contribute to T cell functional suppression in this study[31]. Another recent study reported that NR provision rescued impaired glycolysis in a cellular model of citrin deficiency. Here, a loss-of-function mutation of the mitochondrial aspartate/glutamate transporter alters activity of the malate/aspartate shuttle and urea cycle, leading to decreased NAD/NADH ratios, and impaired glycolysis and mitochondrial OXPHOS. In this study, NR provision increased cellular NAD levels, partially restoring NAD/NADH ratios and leading to a complete restoration of glycolysis. Mitochondrial function was, however, not rescued, which again agrees with findings here that NRK1 activity preferentially supports glycolysis, linked to localised expression and activity within the cytoplasm[32].

Transcription of NAMPT upon T cell activation is controlled by the transcription factors c-Myc and TUB[13]. Since c-Myc expression is promoted by PI3K-Akt signalling, which is required for activation-induced NRK1 expression, these enzymes may therefore share signalling pathways and transcriptional activators driving upregulation upon T cell stimulation. Of note, however, NAMPT inhibition has been reported to more profoundly decrease NAD/H abundance compared to NADP/H[10], the inverse of NRK1 activity demonstrated here, indicating these enzymes may serve complementary roles regulating NAD/H and NADP/H availability, potentially related to subcellular localisation. Indeed, NAMPT is generally reported to be localised within the nucleus, although it may redistribute to the cytoplasm during cell division[33,34]. Recently, NAMPT expression was described to be decreased within tumour-infiltrating T cells, potentially contributing to their well-described metabolic and functional exhaustion. Consistent with this, administration of NAM alongside chimeric antigen receptor (CAR) T cells limited tumour growth and promoted survival in a murine solid tumour model[13]. Whilst the mechanistic basis for decreased NAMPT expression within tumour-infiltrating T cells was not explored, it is plausible that it arises from checkpoint-mediated antagonism of TCR or CD28 signalling to c-Myc. It would be of interest to explore whether NRK1 expression is similarly altered in tumour microenvironments, and to what extent this correlates with T cell metabolic and functional exhaustion.

One notable observation in our study is the increased inflammatory capacity of NRK1-deficient CD4[+] T cells, even in the absence of exogenous NR, in agreement with decreased NAD/H and NADP/H abundance. Different possibilities may explain this. For example, NR could have been generated via dephosphorylation of cellular NMN stores, which is described when hepatocytes are provided NMN[3]. Further support of NR-independent NRK1 activity is provided from studies in yeast, where cellular NAD/H abundance was significantly decreased in NRK1-deleted vs. WT colonies, again in the absence of exogenous NR in either group[35]. Together, these studies identify

potential NRK1 activity independent of dietary NR provision, which our data imply may have a key role in regulating immune cell activity.

Our data also identify that the capacity for cytoplasmic NADP/H synthesis is increased upon T cell activation, with cytoplasmic NADK1 expression being elevated, whilst mitochondrial NADK2 remains unchanged. A recent study reported that the activity of cytoplasmic NADK1, but not mitochondrial NADK2, expands cellular NADP/H pools in cancer cells upon oxidative stress. Conversion of NAD/H to NADP/H was increased by 30–40% under these conditions. In these cells, NADK1 expression remained stable, but rather its activity was increased via direct interaction with G6PD, expression of which was increased during oxidative stress[24]. It is possible that interaction with G6PD further promotes NADK1 activity in CD4[+] T cells, since G6PD transcripts are also increased upon CD4[+] T cell stimulation. Indeed, inhibition of G6PD activity here did elevate ROS abundance and promote inflammatory cytokine expression, but this may more likely be explained by the key role of G6PD for NADP+ reduction to NADPH rather than by interaction-driven increases in NADK1 activity. Indeed, treatment of murine CD8[+] T cells with G6PDi profoundly decreases NADPH/NADP, indicating limited NADPH generation via other routes, such as malic enzyme or isocitrate dehydrogenase 1 in T cells. Again, this was linked to increased ROS abundance, albeit in this case, cytokine expression was suppressed together with cellular viability[36]. Of note, it is reported that T cells express NADPH oxidases, which generate ROS upon activation, impacting signalling pathways to promote pro-inflammatory Th1 differentiation and IFN-γ expression[37–39]. It is possible that NRK1 activity supports this via the generation of upstream NAD/H and NADP/H, albeit the hyper-activated phenotype and increased oxidative damage and cell death of NRK1-deficient cells indicate a more critical role in ROS regulation vs. generation.

In monocytes, NR supplementation, or alternative approaches to increase cellular NAD/H, including CD38 knockdown and PARP/SIRT inhibition, decreased rates of autophagy, associated with increased inosine abundance[6]. Since autophagy importantly supports CD4[+] T cell metabolism, survival, as well as pro-inflammatory Th1 and Th17 differentiation and function, it would be pertinent to investigate whether NRK1 activity modulates autophagy in CD4[+] T cells[40]. Additionally, in that study, increased PPP metabolites were detected in NR-treated monocytes, including ribose-5-phosphate and seduheptolose-7-phosphate, which were decreased in NRK1-deficient T cells studied here, indicating NRK1 activity regulates PPP activity in diverse immune cell subsets[6].

NAD is a co-substrate for both SIRT and PARP enzymes. Decreased SIRT activity was not explored here as a potential mechanistic basis for the observed hyper-functionality of NRK1-deficient CD4[+] T cells, however SIRT1 activity is reported to support pro-inflammatory hybrid Th1/Th17 CD4[+] T cell development, linked to expression of stemness-associated genes, with both inhibition and knockdown impairing effector functions and anti-tumour capacity of these cells[14], which would be inconsistent with observations here that decreased NAD abundance also promotes inflammatory cytokine expression. Direct assessment of PARP activity indicated this was similar in WT and NRK1KO cells and therefore unlikely to underpin the increased inflammatory function of NRK1KO CD4[+] T cells. PARP1 and 2 activity have previously been reported to control T cell differentiation and function[41,42], therefore, the lack of involvement here may relate to expression of NRK1 increasing more in the cytoplasm vs. nucleus upon activation.

Our data identify that NRK1 expression determines maintenance of effector, IFN-γ-expressing CD4[+] T cells within infected tissues during invasive infection with *C. neoformans* and, importantly, contributes to infection control in the brain. CD4[+] T cells are critical for protection from this disease, and clinical outcomes correlate with cerebrospinal CD4[+] T cell frequencies and IFN-γ levels[28,29]. Moreover, IFN-γ treatment demonstrates effectiveness in clinical trials, highlighting potential for

therapeutic intervention to modulate its expression[28]. Despite this, factors controlling protective CD4+ T cell immune responses and IFN-γ expression in this disease are poorly characterised. This work has revealed that targeting NRK1 activity specifically, or CD4+ T cell metabolism and ROS homoeostasis generally, may be an important new area for future study with clear translational potential, indeed little is known about the metabolic control of immune responses in fungal infection, or in the brain.

A key technical advance of this work is the use of two complementary approaches to study NAD/H and NADP/H abundance at the subcellular level, applied for the first time to primary immune cells. Use of these confirmed NRK1 activity is enriched specifically within the cytoplasm of activated CD4+ T cells, and cytoplasmic NAD/H levels generally increase in activated vs. resting cells. This is consistent with altered synthetic enzyme expression patterns (i.e. NMNAT1 vs. NMNAT3) and increased glycolytic capacity of activated T cells, and furthermore indicates localised NAD/H abundance may determine relative activity of metabolic pathways within distinct subcellular compartments at different stages of immune cell differentiation. Further analysis of metabolite abundance and labelling patterns within purified cell fractions would be of interest to determine this, which could be coupled with the use of compartment-specific biosensors upon manipulation of NAD synthesis enzyme expression.

Taken together, the data presented here identify a critical role for NRK1 in regulating CD4+ T cell viability and differentiation by controlling cytoplasmic NAD/H and NADP/H abundance, with implications for ROS homoeostasis and NFAT activity. This provides a mechanistic basis informing potential NR supplementation for immune-mediated and inflammatory diseases and key insight into the biology of NAD synthesis in these cells. Moreover, this work highlights that control of immune cell metabolism at the subcellular level determines whole-organism immune responses.

## Methods

### Healthy volunteer peripheral blood donors
Human peripheral blood mononuclear cells (PBMCs) were isolated from fully anonymized leucocyte cones collected from NHS Blood and Transplant (NHSBT), Birmingham, UK, or healthy volunteer blood donors. All volunteers signed a consent form. All human studies were approved by the University of Birmingham STEM Ethics Committee (Ref. ERN 17_1743).

### Mice
8-12 week old mice (males and females) were housed in individually ventilated cages under specific pathogen-free conditions at the Biomedical Services Unit at the University of Birmingham, and had access to standard chow and drinking water ad libitum. Mice were housed under 12 h light/dark cycle at 20–24 °C and 45–65% humidity. Experiments with transgenic mice utilised both males and females to maintain littermate controls.

Wild type (WT) refers to corresponding littermates. Mice homozygous for the NRK1tm1a allele were purchased from the Wellcome Trust Sanger Institute (MGI ID:4419121) and bred with the FLPer deleter strain (Jackson Laboratories) to remove the FRT-flanked knock-out first cassette, generating NRK1tm1c homozygous mice (referred to as NRK1Flox in this manuscript). Homozygous NRK1Flox animals were bred with heterozygous CD4Cre transgenic mice (kindly provided by Prof. Graham Anderson, University of Birmingham) to generate NRK1Flox/CD4Cre+ and NRK1Flox/CD4Cre- littermate controls. NRK1tm1c mice were also bred with phosphoglycerate kinase 1 (PGK) Cre transgenic mice (Jackson Laboratories) to generate global NRK1KO (tm1d) lines. Mice were euthanised by cervical dislocation at the indicated analysis time-points. All animal studies were approved by the United Kingdom Home Office (Project Licences PP5876109 (SD) and PP4085778 (EWR)).

### Statistical analysis
Data are presented as the mean and individual replicate values. Certain data are presented as normalised values. For human samples, data were normalised within each individual donor by calculating the average (mean) value of all matched samples acquired from that donor, dividing each of the individual values by that mean and multiplying by 100. For murine samples, data were normalised by calculating the average (mean) value of all samples from all mice analysed per batch (balanced for genotype) and then dividing each of the individual values by that mean.

Individual data points were removed if identified as outliers with the ROUT outlier test (one data point in each of Fig. 1B; Supplementary Fig. 1G; Fig. 3K and Supplementary Fig. 7B). For Supplementary Fig. 1G and Fig. 3K this altered the statistical analysis. Paired data were analysed by two-sided paired Student's $t$ test for two conditions, repeated measures ANOVA with Holm-Sidak's post-hoc test for more than two conditions or two-way ANOVA with Holm-Sidak's post-hoc test for more than one parameter. Unpaired data were analysed by two-sided unpaired $t$ test. Specific tests used are indicated in the figure legend. Analysis was performed using GraphPad Prism 9.

### Reporting summary
Further information on research design is available in the Nature Portfolio Reporting Summary linked to this article.

## Data availability
The mass spectrometry data generated in this study have been deposited in the MassIVE database under accession codes MSV000100190 (GC-MS, ftp://massive-ftp.ucsd.edu/v11/MSV000100190) and MSV000100189 (LC-MS, ftp://massive-ftp.ucsd.edu/v11/MSV000100189/). All other data are available upon reasonable request to the corresponding author. Source data are provided as a Source Data file. Source data are provided with this paper.

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

## Acknowledgements

This study was supported by an MRC New Investigator Research Grant to SD (Ref. MR/V011588/1). B.M. and D.A.T. were supported by the award Ref. C42109/A24757 from CRUK. Stable isotope tracing experiments were conducted, with additional analysis support at the Metabolic Tracer Analysis Core at the University of Birmingham and Seahorse analyses at the Cellular Health and Metabolism Core at the University of Birmingham. NAD biosensor and cpVenus plasmid vectors were sourced from Addgene, but with the kind support of Prof X Cambronne. We also thank Prof Graham Anderson for the provision of CD4 Cre mice.

## Author contributions

V.S., M.A., N.G., E.L.B., T.F.-W., B.T., S.H. (Silke Heising), S.H.M., S.H. (Sofia Hain), L.G., J.M., L.S., S.P.D., B.M., M.D.C.L.D., D.A.B., E.W.R., R.A.D. and S.D. developed experimental protocols, undertook experiments and analysed data. V.S., D.A.T., C.L.D., E.W.R., G.G.L., R.A.D., and S.D. oversaw the study, wrote and reviewed the manuscript and sourced financial support.

## Competing interests

DAT undertakes paid consultancy work for Sitryx Ltd. The remaining authors declare no competing interests.
