## [Peer Review File · Nature Communications]

Cytoplasmic NAD/H synthesis via NRK1 regulates inflammatory capacity and promotes survival of CD4+ T cells

Corresponding Author: Dr Sarah Dimeloe

Version 0:

Reviewer comments:

Reviewer #1

(Remarks to the Author)

Stavrou et al. provide a detailed account of how NRK1 supports glycolysis and NADP/H levels in CD4+ T cells. This includes the description of possible mechanisms of how this could alter immune function. However, the impact of NRK1 on the function of CD4+ T cells requires more investigation to support the claims made.

The authors claim a role for NRK1 in CD4+ survival/viability when stimulated. While this was true in CD3/CD28 stimulated and TH1 polarized spleenocytes, this was not seen in Th17 polarized cells (Supp 7D). This was true even though they all had increased cytokine activity. Does TH2 polarization decrease survival of the NRK1-deficient T cells? This should be further explored and discussed.

The authors state that NRK1 activity is required to control infection in vivo, using a murine pulmonary cryptococcosis model and make this a significant finding in their title, abstract and discussion. However, more evidence is needed to support this claim. The authors make a point that loss of NRK1 in CD4+ T cells in primary organs of infection (lungs and brains Fig. 6H) results in fewer IFN γ positive cells and increased DNA damage but not in the spleens or LN. The brain fungal burdens were provided as evidence of a loss of fungal control in NRK1^{Flox+}/CD4^{Cre+} mice compared to Cre⁻ controls. However, the biological relevance of this is questionable. The increase while significant is slight, and at levels far below fatal for cryptococcal meningitis in mice. No lung fungal burdens are provided (although mentioned in the methods (line 921)). As this is a pulmonary model and the T cell response is also critical for pulmonary cryptococcal control, this data should be shown. Showing the same loss of fungal control in the lungs with the same loss of IFN γ positive cells and increased DNA damage seen in the brains but not in the spleens would strengthen the mechanistic argument. As it stands the generalization of the role of NRK1 in inflammatory control of pathogens is premature. A secondary infection model would strengthen the argument of NRK1 activity as fundamental mechanism of inflammatory control.

Reviewer #2

(Remarks to the Author)

Key Findings/Summary

In this manuscript, Stavrou and colleagues are examining the role of NRK1 in T cell activation. NRK1 converts nicotinamide riboside (NR) into nicotinamide mononucleotide (NMN) which is converted into NAD⁺ by nicotinamide mononucleotide adenylyl transferase (NMNATs). The authors demonstrate that supplementation of NR in vitro upregulates NRK1 expression and increases the cellular pool of NAD/H in both human and murine CD4+ T cells. Building on previously published data, the authors demonstrate that NR supplementation results in decreased CD4+ T cell activation.

To demonstrate that the CD4+ T cell phenotype is driven by NRK1 activity and subsequent increases in NAD/H levels, the authors utilized a global *Nrk1* KO mouse model. As expected, KO of *Nrk1* results in hyper-activity of CD4+ T cells, which correlates with increased cellular ROS. It is well established that cellular ROS promotes NFAT activation and nuclear translocation, which is also seen in the current study in *Nrk1* KO cells.

This suggests a mechanism whereby a loss of NRK1 results in decreased NAD/H and NADP/H levels in the cell which

ultimately results in hyperactivation of T cells due to increased ROS and NFAT activation. Ultimately, hyperactivation is not associated with protection from invasive fungal disease after *Cryptococcus neoformans* infection and is instead associated with decreased viability of CD4+ T cells and reduced trafficking of CD4+ T cells to infected organs. Overall, this suggests that NR supplementation may be beneficial in the infectious setting by regulating NFAT levels to ensure optimal activation, trafficking, and function of CD4+ T cells.

Significance

The data presented by Stavrou and colleagues is very thorough and makes a substantial contribution to the field of T-cell biology. The manuscript is greatly strengthened by utilizing multiple approaches to confirm their findings. Given the public and scientific interest in the purported health benefits of NAD+ supplementation, the insight provided by this manuscript into how NAD+ supplementation might affect health in the context of T-cell biology is significant to the field. I have provided some suggestions below that I hope the authors will find useful in strengthening the manuscript for publication.

Methodology

- **Validity of Knockout Model:** Given that the knockout models presented in the manuscript were created in-house, there should be published data that validates the model. If this has been previously published, a citation to that paper will be fine. Can the authors provide sequencing or PCR data that demonstrates the removal of the *Nrk1tm1c* allele? Figure 2A demonstrates a lack of *Nrk1* transcripts via qPCR for the global knockout but there is no data confirming knockout of *Nrk1* in the CD4+-specific KO model and no data confirming that the knockout was only in CD4+ T cells. If an antibody for mouse NRK1 is available, a western blot demonstrating knockout would provide stronger evidence than qPCR, especially without knowing where in the transcript the primers bind relative to the sequence that was removed.
- **Flow Cytometry Gating:** The flow cytometry gating strategies are not explained or provided in full detail. The authors state they used a viability dye in the methods but don't explain whether or not they gated on viable cells before selecting other cell populations. A full sample gating strategy should be provided in the supplemental data showing any steps that were taken to remove doublets, etc. In supplemental figures 2 and 6, it is not clear if the cells in panel b are from the gate drawn in panel A, etc. I assume they are as this is standard practice, but it should be explicitly stated. Also missing are FMO or isotype controls for most flow cytometry data, except for what is presented in Figure 6. In Figure 2 there is no representative plot for IL-2 staining. Importantly, the flow cytometry panels in supplemental figures 2 and 6 have an identical flow plot for panel D even though this is supposed to be a different mouse model.
- **Vehicle Controls:** In supplemental figure 1, several inhibitors are used but it is unclear what the vehicle controls were if any. Panel F shows 0 as one of the concentrations for each inhibitor which is likely a vehicle control, but this is not stated. For panel H there is no vehicle control listed.
- **Control Conditions:** The authors state in line 113 that the CD3/CD28-stimulated cells supplemented with NR were compared to control conditions. Please clearly state what the control conditions are.
- **Infection Outcomes:** The authors demonstrated on day 14 that CD4+-specific *Nrk1* knockout mice have an increased fungal burden in the brain. How does this correlate to outcomes? Do normal mice clear the infection or is it lethal? How does *Nrk1* knockout affect these outcomes? Does it decrease survival time or increase clearance time?
- **Mitochondrial Dependence:** This seems like an unnecessarily complicated value to calculate. Why not just show mitochondrial capacity goes down like glycolytic capacity? This is not my area of expertise so this may be standard practice.

Data Analysis

- **Null hypothesis:** The authors checked N/A in the summary sheet for null hypothesis testing and reporting effect sizes, p-values, and 95% confidence intervals. However, every statistical test used is in fact testing a null hypothesis.
- **Normalization:** One concern is the normalization method utilized by the authors throughout the manuscript. In some experiments, they choose to normalize the data and in others, they do not and show raw data. There is no justification for why normalization is performed in some cases and not in others. To avoid p-hacking (i.e. choosing whichever method produces the smallest p-value), the criteria for determining when to normalize should be selected before acquiring the data and explained. It is also not very clear how they normalized the data. They state that the data is normalized as a percentage of the mean value across all samples analyzed for each donor. Does this mean only untreated? Can you provide a sample calculation? If you are performing paired T tests, normalization should not be necessary as it is designed to calculate the mean of the paired differences rather than calculating the differences in means between the groups.
- **Outliers:** The authors report that outliers were removed whenever flagged by the ROUT test in GraphPad Prism. However, they should also report which datasets had outliers removed. Are there any occasions where the removal of outliers changed the results of the analysis? If so, this should be explicitly stated along with a justification for removing the datapoint.
- **One-Sided or Two-Sided:** The authors do not state if any of their tests are one-sided or two-sided although they checked "Confirmed" on the reporting summary indicating that they did.
- **Missing Stats for No Differences:** In figures where the authors are trying to demonstrate no difference, they don't list statistical tests used for those figures. If there is no difference this should be determined by a statistical test with a high p-value or at least a report of the 95% CI that includes the null hypothesis value for the particular test.
- **Multiple Comparisons:** Figure 2 G shows a time course for cytokine expression and was analyzed by several independent T tests. The authors should correct for multiple comparisons.

Context and Clarity

- In line 65, the authors cite previous literature showing NR supplementation results in decreased circulating cytokine levels but do not specify which cytokines were affected.
- The authors do not state a hypothesis for this study. Was this work driven by a hypothesis or is it exploratory in nature?
- Given the broad audience of this journal, I recommend that the first time NAD/H or NADP/H is mentioned in the text that the authors explain this is referring to the combined pools of NAD⁺ and NADH or NADP⁺ and NADPH, respectively.
- The authors did not mention any possible role for increased NADP/H levels and NADPH oxidase activity. It has been previously shown that T cell activation results in NADPH oxidase activity which promotes ROS production. The authors should discuss their findings in the context of NADPH oxidase as well as metabolic pathways.

Validity of Conclusions

- The authors conclude based on the data presented in Figure 1 that the use of the NAMPT inhibitor confirms that NR activity is mediated by NRK1. I think this wording is a bit confusing and would be better rewritten as "...NR supplementation leads to ND/H levels via NRK1. However, I think this conclusion is a bit overstated and assumes that in the absence of the NAM salvage pathway any increase in NAD/H after NR supplementation must be mediated by NRK1. The reason for the knockout model is to prove that NRK1 is involved, so this conclusion is a bit premature.
- When discussing the pentose phosphate pathway (PPP), the authors state that there was no difference in ribulose-5-phosphate levels (a metabolite dependent on NADP/H). However, they note significant differences in xylulose-5-phosphate and sedoheptulose-7-phosphate levels (metabolites not dependent on NADP/H). While their ultimate conclusion is that the PPP was not significantly impacted, the discussion of the data leads the reader to expect otherwise. To strengthen this section, the authors could explicitly differentiate metabolites dependent on NADP/H from those that are not and better align their discussion with the conclusion.
- The remainder of the conclusions made by the authors are well supported by the data presented.

Minor Changes and Clarifications

- Line 152 "...increases in activity...". This could be clearer by specifying what activity the authors are referring to.
- Line 172 – The authors state that maximal OCR is "slightly" decreased. I suggest the authors avoid the use of the word "slightly" which is qualitative at best. Figure 3C shows there is a statistically significant decrease in OCR activity.
- Lines 192-194 – Missing reference for "...previously observed that in activated T cells, glutamine is a larger contributor..."
- Line 220 – The subjective qualifier "substantially" is open to interpretation. When looking at the data from Figure 4A, the WT cells do have an increase in NAD/H levels when comparing stimulated with and without NR. Is this difference statistically significant?
- There are no in-text callouts for Supplemental Figures 6B-D
- No units for G6PD inhibitor in Figure 4H. No units for Supplemental Figure 1F
- Gate labels in Supplemental Figures 2 and 6 are too small to read. They should be enlarged as was done for the main figures.
- Figure 3D is missing labels for ECAR corresponding to glucose addition, oligomycin addition, etc. Are they the same as the OCR plot? Traditionally, ECAR plots have 2-DG addition, but it doesn't look like that was used here so it would be beneficial to label the figure the same as 3A.
- Figure 5D has the top panel labeled "Mock" but no label for the bottom panel.
- Supplemental Figure 1 legend on line 947 has "D" as a label for summary data when it should be panel "E". The legend also has "or time points indicated" for panel A but no points appear until panel E.

Version 1:

Reviewer comments:

Reviewer #1

(Remarks to the Author)

I thank the authors for their addressing of my comments.

Reviewer #2

(Remarks to the Author)

The revisions made by the authors have addressed a majority of my major concerns outlined in my previous review. The addition of the bone marrow chimera influenza model has greatly strengthened the conclusions in the manuscript. This not only shows a similar phenotype in a different disease context, but the bone marrow chimera approach allows for direct comparison of CD4 T cells with and without NRK1 in the same environment, which clearly demonstrates a difference.

I want to acknowledge the substantial amount of work the authors have put into this manuscript and addressing reviewer comments that I want to acknowledge. There are still a few concerns that have not been adequately addressed that I think could be improved upon to produce a stronger manuscript.

1. The validity of the knockout model - In my first critique, I mentioned that there was no data demonstrating the validity of the knockout mouse. Specifically, the paper lacks data demonstrating that the *Nrk1tm1c* allele was generated after crossing the mouse with the FLP^{er} mouse and subsequently crossing with the CD4Cre and PGKCre mice. The authors should show evidence that crossing with the two Cre mouse models successfully removed the intended exon (exon 2) via PCR or sequencing. While qPCR showing a substantial decrease in *Nrk1* transcripts is also necessary, it cannot stand alone to prove the genetic modifications occurred as intended. For the CD4 KO mice, there should be a negative control showing that the knockout is CD4-specific. In other words, in addition to showing a decrease in *Nrk1* transcripts in CD4 cells, there should be another cell type where there is no difference to demonstrate specificity.

2. Flow Cytometry Gating - My concerns about showing the gating strategies and appropriate controls have been addressed. However, upon further review, I found an additional flow cytometry concern I missed in the first review. In Supplemental Figures 2 and 6, the gating strategy for Tregs does not make sense. Tregs are typically defined as the population that is double positive for FoxP3 and CD25. However, the authors have gated on CD25 single-positive cells as they have included the full range of FoxP3 expression in their gate. This should be corrected, but it does not impact the overall conclusions of the figure or the paper. I apologize for missing this on my first review.

3. Statistics in Figure Legends - The authors added a description of the statistics used in the figure legends as I requested, but some typos/mistakes were introduced with this revision. Specifically, the figure legend for Figure 2 doesn't match the tests with the panels coherently. I recommend double-checking this and the rest of the figure legends.

4. Context and Clarity Concerns - I recommended previously that the authors should explain to the broad audience of this journal that the terms NAD/H or NADP/H are referring to the combined pools of NAD⁺ and NADH or NADP⁺ and NADPH, respectively. This clarification was added to the results section but not to the introduction, where the terms are first mentioned and discussed. I also asked about whether this was a hypothesis-driven study which the authors responded that it was, but this was not added to the manuscript.

Version 3:

Reviewer comments:

Reviewer #2

(Remarks to the Author)

The authors have adequately addressed all of my comments. I am still surprised that the authors did not confirm at the genetic level that their *Nrk1tm1c* x PGK-cre crossing was successful. While looking at the transcript via qPCR is a useful screening method, a new mouse line should always be confirmed at the genetic level.

However, since the authors' conclusions are not solely based on this single model, but rather on multiple other models, such as human cells and CD4-specific *Nrk1* KO, the findings of this paper remain valid in my opinion.

We thank the reviewers for their time and attention spent reviewing our manuscript and for the helpful suggestions to further strengthen the findings and improve its clarity.

Reviewer #1 (Remarks to the Author)

Stavrou et al. provide a detailed account of how NRK1 supports glycolysis and NADP/H levels in CD4⁺ T cells. This includes the description of possible mechanisms of how this could alter immune function. However, the impact of NRK1 on the function of CD4⁺ T cells requires more investigation to support the claims made.

The authors claim a role for NRK1 in CD4⁺ survival/viability when stimulated. While this was true in CD3/CD28 stimulated and TH1 polarized splenocytes, this was not seen in Th17 polarized cells (Supp 7D). This was true even though they all had increased cytokine activity. Does TH2 polarization decrease survival of the NRK1-deficient T cells? This should be further explored and discussed.

Thank you for this suggestion. We have now studied Th2 polarisation in NRK1-deficient and sufficient CD4⁺ T cells. Specifically, as for Th1 and Th17 experiments, whole splenocytes from T cell-specific NRK1KO or control mice (NRK1Flox⁺/CD4Cre^{+/-}) were cultured with anti-CD3/CD28 stimulation and Th2 polarising factors. This promoted generation of CD4⁺ T cells expressing the hallmark transcription factor GATA3 and not IFN- γ , which were not observed under control Th0 conditions (**Supp. Figure 7E-G**). The frequency of this population was similar in cultures containing NRK1-sufficient and -deficient T cells, and viability of CD4⁺ T cells under these conditions was also comparable between genotypes. Therefore, NRK1-deficient CD4⁺ T cells have a greater propensity to differentiate into Th1 and Th17, but not Th2 cells and demonstrate poor viability under Th1 polarising conditions.

The authors state that NRK1 activity is required to control infection in vivo, using a murine pulmonary cryptococcosis model and make this a significant finding in their title, abstract and discussion. However, more evidence is needed to support this claim. The authors make a point that loss of NRK1 in CD4⁺ T cells in primary organs of infection (lungs and brains Fig. 6H) results in fewer IFN γ positive cells and increased DNA damage but not in the spleens or LN. The brain fungal burdens were provided as evidence of a loss of fungal control in NRK1Flox⁺/CD4Cre⁺ mice compared to Cre⁻ controls. However, the biological relevance of this is questionable. The increase while significant is slight, and at levels far below fatal for cryptococcal meningitis in mice. No lung fungal burdens are provided (although mentioned in the methods (line 921)). As this is a pulmonary model and the T cell response is also critical for pulmonary cryptococcal control, this data should be shown. Showing the same loss of fungal control in the lungs with the same loss of IFN γ positive cells and increased DNA damage seen in the brains but not in the spleens would strengthen the mechanistic argument. As it stands the generalization of the role of NRK1 in inflammatory control of pathogens is premature. A secondary infection model would strengthen the argument of NRK1 activity as fundamental mechanism of inflammatory control.

We agree with the reviewer that it would provide a more complete picture of disease during *Cryptococcus neoformans* infection to report lung fungal burdens and have included these in **Fig. 6L**. These data identify no loss of fungal control in the airway in T cell-specific NRK1 deficiency, despite decreased effector CD4⁺ T cell frequencies. Our interpretation of this is that fungal control mechanisms are likely to be tissue/organ specific. This is supported by observations that Cryptococcus-related Immune Reconstitution Inflammatory Syndrome (IRIS), considered to be driven by CD4⁺ T cells, predominantly affects the brain but not the lungs, highlighting heightened CD4⁺ T cell activity in the brain in this infection (reviewed in Wiesner et al, PMID: 22389746). We have revised the text of the results and discussion to clarify and discuss this tissue-specific loss of fungal control.

We also agree with the reviewer that interrogating the role of T cell NRK1 in a secondary infection model would provide further insight and confirmation of its role during immune responses to infection more generally. To this end, we worked with collaborators to assess this in context of a distinct infection context, specifically during *Influenza A* virus infection. To do so, murine bone marrow chimaeras were generated with NRK1Flox⁺/CD4Cre⁺ or NRK1Flox⁺/CD4Cre⁻ bone marrow and intranasally infected with *Influenza A* virus (NRK1Flox⁺/CD4Cre^{+/-} mice could not be transferred between institutional animal facilities).

After 9 days of infection, mice were assessed for effector CD4⁺ T cell frequency within the lung and draining mediastinal lymph node (medLN). Mice were also weighed and assessed for overall disease score daily. Comparison of NRK1Flox⁺/CD4Cre⁺ chimaera mice with NRK1Flox⁺/CD4Cre⁻ controls identified fewer viable, donor-derived CD4⁺ T cells in the lung and medLN at day 9. However, this was also observed in peripheral blood samples taken before infection, indicating poorer engraftment of NRK1-deficient vs. -sufficient CD4⁺ T cells, potentially related to a role for NRK1 during lymphopenia-induced CD4⁺ T cell proliferation (**Supp. Fig 8D-E**), consistent with our observations that this involves increased glucose metabolism (Bachoo et al, PMID 39799568).

To control for this, ratios of donor/host effector CD4⁺ T cells were assessed within individual tissues, which identified consistently fewer donor CD4⁺ T cells expressing IFN- γ , IL-2 and TNF- α in medLN of NRK1Flox⁺/CD4Cre⁺ chimaeric mice compared to NRK1Flox⁺/CD4Cre⁻ chimaera controls (**Supp. Fig 8D, Fig. 6N**), although this was not observed in the lung (**Supp. Fig. 8G**), despite consistently elevated DNA damage (γ H2AX abundance) in CD4⁺ T cells in this site (**Fig. 6O**). DNA damage was not assessed in medLN cells, since the small amount of tissue was prioritised for T cell functional analysis.

In addition, assessment of mouse weight confirmed expected decreases during infection, which were slightly greater in NRK1Flox⁺/CD4Cre⁺ chimaeras, albeit variable, particularly in the NRK1Flox⁺/CD4Cre⁻ group (**Fig. 6P**). Finally, blinded assessment of disease score identified this was greater in NRK1Flox⁺/CD4Cre⁺ chimaeras, particularly at day 8 of infection (**Fig. 6Q**), consistent with reduced effector cell frequencies in draining medLNs.

These new data provide further evidence that NRK1 deficiency impacts CD4⁺ T cell responses to infection *in vivo* with implications for disease control. In addition, since this was observed in a distinct pathogen context it underscores the role for NRK1 in T cell responses to infection more broadly.

Reviewer #3 (Remarks to the Author)

Key Findings/Summary

In this manuscript, Stavrou and colleagues are examining the role of NRK1 in T cell activation. NRK1 converts nicotinamide riboside (NR) into nicotinamide mononucleotide (NMN) which is converted into NAD⁺ by nicotinamide mononucleotide adenylyl transferase (NMNATs). The authors demonstrate that supplementation of NR *in vitro* upregulates NRK1 expression and increases the cellular pool of NAD/H in both human and murine CD4⁺ T cells. Building on previously published data, the authors demonstrate that NR supplementation results in decreased CD4⁺ T cell activation.

To demonstrate that the CD4⁺ T cell phenotype is driven by NRK1 activity and subsequent increases in NAD/H levels, the authors utilized a global *Nrk1* KO mouse model. As expected, KO of *Nrk1* results in hyper-activity of CD4⁺ T cells, which correlates with increased cellular ROS. It is well established that cellular ROS promotes NFAT activation and nuclear translocation, which is also seen in the current study in *Nrk1* KO cells.

This suggests a mechanism whereby a loss of NRK1 results in decreased NAD/H and NADP/H levels in the cell which ultimately results in hyperactivation of T cells due to increased ROS and NFAT activation. Ultimately, hyperactivation is not associated with protection from invasive fungal disease after *Cryptococcus neoformans* infection and is instead associated with decreased viability of CD4⁺ T cells and reduced trafficking of CD4⁺ T cells to infected organs. Overall, this suggests that NR supplementation may be beneficial in the infectious setting by regulating NFAT levels to ensure optimal activation, trafficking, and function of CD4⁺ T cells.

Significance

The data presented by Stavrou and colleagues is very thorough and makes a substantial contribution to the field of T-cell biology. The manuscript is greatly strengthened by utilizing multiple approaches to confirm

their findings. Given the public and scientific interest in the purported health benefits of NAD⁺ supplementation, the insight provided by this manuscript into how NAD⁺ supplementation might affect health in the context of T-cell biology is significant to the field. I have provided some suggestions below that I hope the authors will find useful in strengthening the manuscript for publication.

Methodology

- **Validity of Knockout Model:** Given that the knockout models presented in the manuscript were created in-house, there should be published data that validates the model. If this has been previously published, a citation to that paper will be fine. Can the authors provide sequencing or PCR data that demonstrates the removal of the *Nrk1tm1c* allele? Figure 2A demonstrates a lack of *Nrk1* transcripts via qPCR for the global knockout but there is no data confirming knockout of *Nrk1* in the CD4⁺-specific KO model and no data confirming that the knockout was only in CD4⁺ T cells. If an antibody for mouse NRK1 is available, a western blot demonstrating knockout would provide stronger evidence than qPCR, especially without knowing where in the transcript the primers bind relative to the sequence that was removed.

We have now added qPCR data confirming substantially lower NRK1 transcript levels in CD4⁺ T cells from T cell-specific NRK1-deficient mice. Unfortunately, we do not have a reliable antibody to also confirm this at protein level. Please refer to the new **Figure 6A**.

- **Flow Cytometry Gating:** The flow cytometry gating strategies are not explained or provided in full detail. The authors state they used a viability dye in the methods but don't explain whether or not they gated on viable cells before selecting other cell populations. A full sample gating strategy should be provided in the supplemental data showing any steps that were taken to remove doublets, etc. In supplemental figures 2 and 6, it is not clear if the cells in panel b are from the gate drawn in panel A, etc. I assume they are as this is standard practice, but it should be explicitly stated. Also missing are FMO or isotype controls for most flow cytometry data, except for what is presented in Figure 6. In Figure 2 there is no representative plot for IL-2 staining. Importantly, the flow cytometry panels in supplemental figures 2 and 6 have an identical flow plot for panel D even though this is supposed to be a different mouse model.

All gating strategies are now provided in full and referred to at the relevant places within the results sections. Arrows have also been added to the gating strategies to clarify the gating hierarchy. FMO controls are now shown for CD25, PD-1 (**Supp. Fig. 2I**), DCFDA, NFAT (**Supp. Fig. 4A,C**), γ H2AX (**Supp. Fig. 7J**), IFN- γ , TNF- α and IL-2 (**Supp. Fig. 8F**). Secondary antibody only controls are shown for NRK1 (**Figure 1, Supp. Fig. 1F**).

We have also added a representative plot for IL-2 staining to Figure 2 and replaced the incorrectly duplicated panels in Supp. Fig. 6 – we thank the reviewer for highlighting this error.

- **Vehicle Controls:** In supplemental figure 1, several inhibitors are used but it is unclear what the vehicle controls were if any. Panel F shows 0 as one of the concentrations for each inhibitor which is likely a vehicle control, but this is not stated. For panel H there is no vehicle control listed.

Thank you, we have added further details on the vehicle control to the figure legend and some additional clarification within the methods. The figure legend for this panel now reads "(F) Human CD4⁺ T cells were stimulated via CD3/CD28 for 48 hours in presence of indicated inhibitors (or DMSO vehicle control) and assessed for NRK1 protein abundance by flow cytometry (representative histograms and summarised for n=3 independent donors)". In the methods we have added "Vehicle controls were added equivalent to the highest volume of additions".

- **Control Conditions:** The authors state in line 113 that the CD3/CD28-stimulated cells supplemented with NR were compared to control conditions. Please clearly state what the control conditions are.

We have now amended this sentence for clarity. It now reads "We first assessed activation status upon CD3/CD28 stimulation in presence of NR, compared to untreated control cells."

- Infection Outcomes: The authors demonstrated on day 14 that CD4⁺-specific *Nrk1* knockout mice have an increased fungal burden in the brain. How does this correlate to outcomes? Do normal mice clear the infection or is it lethal? How does *Nrk1* knockout affect these outcomes? Does it decrease survival time or increase clearance time?

In the intranasal infection model for *Cryptococcus neoformans*, which best recapitulates human exposure, susceptibility is driven by overwhelming lung infection which would tend to kill mice before significant brain burdens can develop. However, animal research licenses in the UK do not generally allow the model to extend to this point, but rather define humane end-points for euthanising mice, defined by weight loss (maximum 15%) in combination with other clinical signs of infection.

Because of the initial overwhelming lung infection, it is not possible to model fatal levels of brain infection with intranasal infection, which instead would require intravenous administration (after which the pathogen directly infects the brain rather than disseminating from the lung). However, such experiments would be out of scope of the current study, as it would significantly change how CD4⁺ T cells behave and infiltrate the brain (the subject of an ongoing study in revision for this journal, by our collaborator Dr Drummond).

- Mitochondrial Dependence: This seems like an unnecessarily complicated value to calculate. Why not just show mitochondrial capacity goes down like glycolytic capacity? This is not my area of expertise so this may be standard practice.

These parameters are standard for this assay (Arguello et al, PMID 33264598) and, I believe, relate to the fact that mitochondrial inhibitors force cells to rely on glycolysis for ATP generation (and ATP-dependent puromycin incorporation, the readout of the assay), allowing calculation of glycolytic capacity, whereas upon inhibition of glycolysis, several pathways upstream of mitochondrial activity may compensate (e.g. fatty acid oxidation / amino acid catabolism), in addition to increased mitochondrial function *per se*, making it challenging to similarly define mitochondrial capacity.

Data Analysis

- Null hypothesis: The authors checked N/A in the summary sheet for null hypothesis testing and reporting effect sizes, p-values, and 95% confidence intervals. However, every statistical test used is in fact testing a null hypothesis.

We thank the reviewer for this clarification.

- Normalization: One concern is the normalization method utilized by the authors throughout the manuscript. In some experiments, they choose to normalize the data and in others, they do not and show raw data. There is no justification for why normalization is performed in some cases and not in others. To avoid p-hacking (i.e. choosing whichever method produces the smallest p-value), the criteria for determining when to normalize should be selected before acquiring the data and explained. It is also not very clear how they normalized the data. They state that the data is normalized as a percentage of the mean value across all samples analyzed for each donor. Does this mean only untreated? Can you provide a sample calculation? If you are performing paired T tests, normalization should not be necessary as it is designed to calculate the mean of the paired differences rather than calculating the differences in means between the groups.

We thank the reviewer for highlighting this was not clear. Data were normalised to control for differences between individual experiments performed at different times and permit overall patterns to be more clearly defined. This was applied to mean fluorescence intensity (MFI) values from flow cytometry experiments, which can vary between experiments because of alterations to instrument settings and is our standard approach for this type of data. Normalisation was generally not applied to “percentage positive” data from flow cytometry experiments (i.e. percentage of cells expressing a certain marker or cytokine) since such data are based on gating strategies with negative controls and are more stable over time, irrespective of absolute changes in fluorescence signal intensity. Normalisation was also applied to data compiled from repeated luminescent NAD/H or NADP/H assays, again because of inter-assay variability in absolute luminescent signal measured.

For human samples, data were normalised within each individual donor by first calculating the average (mean) value of all matched samples acquired from each donor (including all untreated and treated samples), dividing each of the individual values by that mean and multiplying by 100.

For example:

	Condition 1	Condition 2	Condition 3	Condition 4	Mean
Raw MFI	3567	2452	1457	1206	2170.5
Normalisation	$3567/2170.5*100$	$2452/2170.5*100$	$1457/2170.5*100$	$1206/2170.5*100$	
Normalised MFI	164.3400138	112.9693619	67.12739	55.56323428	

For murine samples, to permit resolution of differences related to genotype, data were normalised by first calculating the average (mean) value of all samples from all mice analysed per batch (which was balanced for KO/WT genotype) and then dividing each of the individual values by that mean.

For example (shown here for one individual animal of each genotype, but experiments were typically undertaken with larger groups/litters):

	Condition 1	Condition 2	Condition 3	Condition 4	Mean
Raw MFI KO	6567	5436	4423	3324	3776.625
Raw MFI WT	3564	2678	2342	1879	
Normalisation KO	$6567/3776.625*100$	$5436/3776.625*100$	$4423/3776.625*100$	$3324/3776.625*100$	
Normalisation WT	$3564/3776.625*100$	$2678/3776.625*100$	$2342/3776.625*100$	$1879/3776.625*100$	
Normalised MFI KO	173.8854136	143.9380399	117.1151491	88.01509284	
Normalised MFI WT	94.36997319	70.90987323	62.01304074	49.7534174	

We have now further clarified this approach in the materials and methods section.

Exceptions to the approach described above include certain instances where flow cytometry MFI or NAD(P)H assay RLU values are shown as raw (not normalised) data, since they were acquired on the same day for several biological replicate samples, with no inter-experimental variability (but these can also be normalised for consistency if helpful). In addition, in Figure 6I, flow cytometry data of “percent positive” cells have been normalised (approach as described above) since there was considerable variability in data between experiments (e.g. range of IFN γ CD4 $^+$ T cells between 0.8 and 8.3%), which may be expected for this *in vivo* infection model.

- Outliers: The authors report that outliers were removed whenever flagged by the ROUT test in GraphPad Prism. However, they should also report which datasets had outliers removed. Are there any occasions where the removal of outliers changed the results of the analysis? If so, this should be explicitly stated along with a justification for removing the datapoint.

We have added a sentence to the methods to provide further detail on which datasets had outliers removed, together with justification and a description of whether this changed the results of the analysis. This sentence reads as follows:

“Individual data points were removed if identified as outliers with the ROUT outlier test (one data point in each of Figure 1B, Supp. Fig 1G, Figure 3K and Supp. Fig 7C). For Supp. Fig 1G and Figure 3K, this altered the statistical analysis.”

- One-Sided or Two-Sided: The authors do not state if any of their tests are one-sided or two-sided although they checked “Confirmed” on the reporting summary indicating that they did.

All tests are two-sided, which has now been clarified in the methods section.

- Missing Stats for No Differences: In figures where the authors are trying to demonstrate no difference, they don’t list statistical tests used for those figures. If there is no difference this should be determined by a

statistical test with a high p-value or at least a report of the 95% CI that includes the null hypothesis value for the particular test.

Details of all statistical tests used have now been included within the figure legends, including for all figures where no significant difference is reported.

- Multiple Comparisons: Figure 2 G shows a time course for cytokine expression and was analyzed by several independent T tests. The authors should correct for multiple comparisons.

Thank you, this has been addressed in the figure and results section.

Context and Clarity

- In line 65, the authors cite previous literature showing NR supplementation results in decreased circulating cytokine levels but do not specify which cytokines were affected.

Thank you, this detail has been added. The sentence now reads: "For example, NR supplementation in healthy volunteers led to decreases in circulating levels of cytokines including interleukin (IL)-2, IL-5, IL-6 and tumour necrosis factor-alpha (TNF- α),⁵"

- The authors do not state a hypothesis for this study. Was this work driven by a hypothesis or is it exploratory in nature?

The study tested the hypothesis that NR, via NRK1 activity, has potential to modulate T cell immune function.

- Given the broad audience of this journal, I recommend that the first time NAD/H or NADP/H is mentioned in the text that the authors explain this is referring to the combined pools of NAD⁺ and NADH or NADP⁺ and NADPH, respectively.

We thank the reviewer for this suggestion, which we have now implemented – please see revised manuscript.

- The authors did not mention any possible role for increased NADP/H levels and NADPH oxidase activity. It has been previously shown that T cell activation results in NADPH oxidase activity which promotes ROS production. The authors should discuss their findings in the context of NADPH oxidase as well as metabolic pathways.

Thank you for this suggestion. We have added some discussion of this to the manuscript. Specifically, we have added "Of note, it is reported that T cells express NADPH oxidases, which generate ROS upon activation, impacting signalling pathways to promote pro-inflammatory Th1 differentiation and IFN- γ expression^{36–38}. It is possible that NRK1 activity supports this via generation of upstream NAD/H and NADP/H, albeit the hyper-activated phenotype and increased oxidative damage and cell death of NRK1-deficient cells indicate a more critical role in ROS regulation vs. generation."

Validity of Conclusions

- The authors conclude based on the data presented in Figure 1 that the use of the NAMPT inhibitor confirms that NR activity is mediated by NRK1. I think this wording is a bit confusing and would be better rewritten as "...NR supplementation leads to ND/H levels via NRK1. However, I think this conclusion is a bit overstated and assumes that in the absence of the NAM salvage pathway any increase in NAD/H after NR supplementation must be mediated by NRK1. The reason for the knockout model is to prove that NRK1 is involved, so this conclusion is a bit premature.

We thank the reviewer for this suggestion and have revised the text as follows "NR treatment also rescued NAD/H levels in presence of the NAMPT inhibitor, FK866 (NAMPTi). In these experiments, failure of NAM

to restore NAD/H in presence of FK866 confirmed effective enzyme inhibition and indicated that NR supplementation increases NAD/H levels via NRK1 (**Supp. Fig. 1J**).”

- When discussing the pentose phosphate pathway (PPP), the authors state that there was no difference in ribulose-5-phosphate levels (a metabolite dependent on NADP/H). However, they note significant differences in xylulose-5-phosphate and sedoheptulose-7-phosphate levels (metabolites not dependent on NADP/H). While their ultimate conclusion is that the PPP was not significantly impacted, the discussion of the data leads the reader to expect otherwise. To strengthen this section, the authors could explicitly differentiate metabolites dependent on NADP/H from those that are not and better align their discussion with the conclusion.

Thank you for this helpful suggestion. We have amended the text, which now reads as follows:

“WT and NRK1KO CD4+ T cells were cultured with 1,2-¹³C-glucose for 24 hours and analysed for intracellular metabolite labelling. This identified decreased abundance of ¹³C-labeled isotopomers of the PPP intermediates xyulose-5-phosphate and seduheptulose-7-phosphate in NRK1KO cells, indicating decreased PPP activity, albeit labelled upstream ribulose-5-phosphate abundance (generation of which is also coupled to NADP+ reduction) was similar (**Fig. 4G**). Further analysis of ¹³C-labeleing of TCA cycle intermediates in these samples identified increased abundance of M+2 isotopomers in NRK1KO vs. WT cells, consistent with earlier observations of increased glucose oxidation (**Supp. Fig 4E**). However, this was not accompanied by increased abundance of M+1 isotopomers, which arise from PPP intermediates re-entering into glycolysis. These data confirm that PPP activity does not increase to the same extent as glucose oxidation in NRK1KO cells, providing further functional evidence of decreased NADP/H abundance and activity.”

- The remainder of the conclusions made by the authors are well supported by the data presented.

Minor Changes and Clarifications

- Line 152 “...increases in activity...”. This could be clearer by specifying what activity the authors are referring to.

This has been revised to “increases in cytokine expression”

- Line 172 – The authors state that maximal OCR is “slightly” decreased. I suggest the authors avoid the use of the word “slightly” which is qualitative at best. Figure 3C shows there is a statistically significant decrease in OCR activity.

Thank you, the word “slightly” has been removed from this phrase

- Lines 192-194 – Missing reference for “...previously observed that in activated T cells, glutamine is a larger contributor...”

Thank you, this has been added.

- Line 220 – The subjective qualifier “substantially” is open to interpretation. When looking at the data from Figure 4A, the WT cells do have an increase in NAD/H levels when comparing stimulated with and without NR. Is this difference statistically significant?

We thank the reviewer for highlighting this point. We have removed the word “substantially”. Since the difference is statistically significant, we have also added an indicator of this to the figure.

- There are no in-text callouts for Supplemental Figures 6B-D

These have been added.

- No units for G6PD inhibitor in Figure 4H. No units for Supplemental Figure 1F

The units for the G6PD inhibitor (nM) have been added to the figure legend for Figure 4.

Units for Supp. Figure 1F have been added to the figure itself for clarity.

- Gate labels in Supplemental Figures 2 and 6 are too small to read. They should be enlarged as was done for the main figures.

Thank you, these have been made larger

- Figure 3D is missing labels for ECAR corresponding to glucose addition, oligomycin addition, etc. Are they the same as the OCR plot? Traditionally, ECAR plots have 2-DG addition, but it doesn't look like that was used here so it would be beneficial to label the figure the same as 3A.

These have been added.

- Figure 5D has the top panel labeled "Mock" but no label for the bottom panel.

Thank you, the transfected label has been added.

- Supplemental Figure 1 legend on line 947 has "D" as a label for summary data when it should be panel "E". The legend also has "or time points indicated" for panel A but no points appear until panel E.

Thank you, these have been corrected.

We thank the reviewers for their time and attention spent reviewing our manuscript and for the helpful suggestions to further strengthen the findings and improve its clarity.

Reviewer #1 (Remarks to the Author):

I thank the authors for their addressing of my comments.

Reviewer #2 (Remarks to the Author):

The revisions made by the authors have addressed a majority of my major concerns outlined in my previous review. The addition of the bone marrow chimera influenza model has greatly strengthened the conclusions in the manuscript. This not only shows a similar phenotype in a different disease context, but the bone marrow chimera approach allows for direct comparison of CD4 T cells with and without NRK1 in the same environment, which clearly demonstrates a difference.

I want to acknowledge the substantial amount of work the authors have put into this manuscript and addressing reviewer comments that I want to acknowledge. There are still a few concerns that have not been adequately addressed that I think could be improved upon to produce a stronger manuscript.

We are happy that the revisions made so far address most of the reviewer's concerns and have greatly strengthened the manuscript.

1. The validity of the knockout model - In my first critique, I mentioned that there was no data demonstrating the validity of the knockout mouse. Specifically, the paper lacks data demonstrating that the *Nrk1^{tm1c}* allele was generated after crossing the mouse with the FLP mouse and subsequently crossing with the CD4Cre and PGKCre mice. The authors should show evidence that crossing with the two Cre mouse models successfully removed the intended exon (exon 2) via PCR or sequencing. While qPCR showing a substantial decrease in *Nrk1* transcripts is also necessary, it cannot stand alone to prove the genetic modifications occurred as intended. For the CD4 KO mice, there should be a negative control showing that the knockout is CD4-specific. In other words, in addition to showing a decrease in *Nrk1* transcripts in CD4 cells, there should be another cell type where there is no difference to demonstrate specificity.

Thank you for this suggestion. We have now undertaken two additional sets of analysis to confirm the validity of the knockout model.

Firstly, we have performed quantitative PCR on DNA isolated from purified T cell populations from NRK1Flox⁺/CD4Cre⁻ and NRK1Flox⁺/CD4Cre⁺ mice, to confirm removal of the intended exons of the *Nrk1* gene. These are exons 5 and 6, as detailed here: <https://www.informatics.jax.org/allele/MGI:4419121> Primer sequences have been added to the Supplemental Methods (new **Table S3**) and a diagram indicating mapping to the relevant exons added to **Supplementary Figure 6A**. T cell DNA was analysed alongside DNA from purified CD19⁺ B cells from the same mice, to additionally confirm cell specificity of the exon deletion. Template abundance was compared to that of a region of genomic DNA in *Ii2ra*, as a control, untargeted reference, using published primers (Jennings et al, STAR protocols, 2021, DOI: 10.1016/j.xpro.2020.100284). The experiments confirmed substantially reduced template abundance in T cells from NRK1Flox⁺/CD4Cre⁺ mice, compared to T cells from NRK1Flox⁺/CD4Cre⁻ mice, as well as B cells from both NRK1Flox⁺/CD4Cre⁺ and

NRK1Flox+/CD4Cre- mice. Residual template likely reflects minor cell sorting impurities. These data have been added to **Supplementary Figure 6B**.

We acknowledge the reviewer's request to perform similar analysis in cells isolated from global NRK1-deficient mice (where the same NRK1-floxed allele was employed together with a PGKCre to delete NRK1 from all cells), however we no longer maintain this mouse colony and do not have material available to undertake such analysis. Since the same NRK1-floxed allele was used as for the CD4Cre-driven model however, we would argue that the data provided confirm validity of this approach to manipulate NRK1 expression, further supported by analysis of mRNA transcript abundance in the NRK1KO model.

Secondly, as further confirmation of the cell specificity of NRK1 manipulation in the CD4Cre-driven model, we undertook qPCR analysis of mRNA transcript abundance in purified splenic CD19+ B cells from NRK1Flox+/CD4Cre+ and NRK1Flox+/CD4Cre- mice, stimulated for 24 hours with LPS. This analysis indicated similar transcript abundance in both groups. This data has now also been added to **Supplementary Figure 6H** and method details also added.

2. Flow Cytometry Gating - My concerns about showing the gating strategies and appropriate controls have been addressed. However, upon further review, I found an additional flow cytometry concern I missed in the first review. In Supplemental Figures 2 and 6, the gating strategy for Tregs does not make sense. Tregs are typically defined as the population that is double positive for FoxP3 and CD25. However, the authors have gated on CD25 single-positive cells as they have included the full range of FoxP3 expression in their gate. This should be corrected, but it does not impact the overall conclusions of the figure or the paper. I apologize for missing this on my first review.

Thank you. We have now adjusted this gating strategy. Please see the revised Supplementary Figures 2 and 6 for changes.

3. Statistics in Figure Legends - The authors added a description of the statistics used in the figure legends as I requested, but some typos/mistakes were introduced with this revision. Specifically, the figure legend for Figure 2 doesn't match the tests with the panels coherently. I recommend double-checking this and the rest of the figure legends.

Thank you. We have now corrected this error and checked all other legends.

4. Context and Clarity Concerns - I recommended previously that the authors should explain to the broad audience of this journal that the terms NAD/H or NADP/H are referring to the combined pools of NAD+ and NADH or NADP+ and NADPH, respectively. This clarification was added to the results section but not to the introduction, where the terms are first mentioned and discussed. I also asked about whether this was a hypothesis-driven study which the authors responded that it was, but this was not added to the manuscript.

Thank you. We have now addressed these points. Please see the revised introduction section.